# Addressing the difficulties in quantifying droplet number response to aerosol from satellite observations

Hailing Jia[1], Johannes Quaas[1], Edward Gryspeerdt[2,3], Christoph Böhm[4], and Odran Sourdeval[5]

[1]Leipzig Institute for Meteorology, Universität Leipzig, Leipzig, Germany
[2]Space and Atmospheric Physics Group, Imperial College London, UK
[3]Grantham Institute for Climate Change and the Environment, Imperial College London, UK
[4]Institute for Geophysics and Meteorology, University of Cologne, Cologne, Germany
[5]Laboratoire d'Optique Atmosphérique, Université de Lille, CNRS, Lille, France

**Correspondence:** Hailing Jia (hailing.jia@uni-leipzig.de)

**Abstract.** Aerosol–cloud interaction is the most uncertain component of the overall anthropogenic forcing of the climate, in which cloud droplet number concentration ($N_d$)-to-aerosol sensitivity ($S$) is a key term for the overall estimation. However, satellite-based estimates of $S$ are especially challenging, mainly due to the difficulty in disentangling aerosol effects on $N_d$ from possible confounders. By combining multiple satellite observations and reanalysis, this study investigates the impacts of a) updraft, b) precipitation, c) retrieval errors, as well as d) vertical co-location between aerosol and cloud, on the assessment of $S$ in the context of marine warm (liquid) clouds. Our analysis suggests that $S$ increases remarkably with both cloud base height and cloud geometric thickness (proxies for vertical velocity at cloud base), consistent with stronger aerosol-cloud interactions at larger updraft velocity for mid- and low-latitude clouds. In turn, introducing the confounding effect of aerosol–precipitation interaction can artificially amplify $S$ by an estimated 21 %, highlighting the necessity of removing precipitating clouds from analyses on $S$. It is noted that the retrieval biases in aerosol and cloud appear to underestimate $S$, in which cloud fraction acts as a key modulator, making it practically difficult to balance the accuracies of aerosol–cloud retrievals at aggregate scales (e.g., $1° \times 1°$ grid). Moreover, we show that using column-integrated sulfate mass concentration (SO4C) to approximate sulfate concentration at cloud base (SO4B) can result in a degradation of correlation with $N_d$, along with a nearly twofold enhancement of $S$, mostly attributed to the inability of SO4C to capture the full spatio-temporal variability of SO4B. These findings point to several potential ways forward to account for the major influential factors practically by means of satellite observations and reanalysis, aiming at optimal observational estimates of global radiative forcings due to the Twomey effect and also cloud adjustments.

## 1 Introduction

Aerosol particles, by acting as cloud condensation nuclei (CCN), can modify cloud properties and precipitation formation, altering the radiative flux at the top-of-atmosphere, which is known as effective radiative forcing from aerosol-cloud interactions ($ERF_{aci}$) (Forster et al., 2021). Additionally, absorbing aerosols can also alter the cloud distribution by perturbing the atmospheric temperature structure, known as semi-direct effects (Allen et al., 2019). $ERF_{aci}$ may be further subdivided into (i)

the radiative forcing due to aerosol-cloud interactions (RF$_{aci}$), also known as the Twomey effect, describing the increased cloud albedo resulted from enhancement in cloud droplet number concentration ($N_d$) due to an increase in anthropogenic aerosol emissions (Twomey, 1974), and (ii) rapid adjustments, which are essentially the consequent responses of liquid water path and cloud horizontal extent to changed $N_d$ via the Twomey effect (Albrecht, 1989; Ackerman et al., 2004; Zhao and Garrett, 2015; Bellouin et al., 2020). Although extensive investigations have been made to quantify the Twomey effect, significant uncertainties remain on its magnitude. This study will discuss the Twomey effect with a focus on the sensitivity of $N_d$ to CCN perturbations, due to its fundamental role in aerosol-cloud interactions. Note that the related radiative forcing will be not estimated here, as the anthropogenic perturbation to CCN concentrations is highly uncertain and not easily accessible from observational data.

Current climate models suggest diverse magnitudes of the Twomey effect even with identical anthropogenic aerosol emission perturbation (Gryspeerdt et al., 2020; Smith et al., 2020). Thus, observational data at the climate-relevant scale, i.e., satellite retrievals, are required to quantify and constrain the Twomey effect globally, basically the sensitivity of $N_d$ to CCN perturbations (Seinfeld et al., 2016). As reviewed recently by Quaas et al. (2020), there are, however, several uncertainties in inferring the $N_d$-to-CCN sensitivity ($S=\frac{d\ln N_d}{d\ln N_{CCN}}$, where $N_{CCN}$ means proxies for CCN number concentration) from satellite observations, hindering its applicability to further evaluate climate models or quantify RF$_{aci}$ from data. Most of them have been reported to bias $S$ toward a lower value, in turn leading to an overall underestimated ERF$_{aci}$, including (i) the instrument detectability limitations on aerosol loading in pristine environments (Ma et al., 2018a), (ii) the inadequate proxy (such as aerosol optical depth (AOD) or a variant thereof) for CCN owing to the lack of information on the aerosol size and chemical composition (Stier, 2016; Hasekamp et al., 2019), (iii) the limited usability of AOD–$N_d$ relationship under present day (PD) to determine the change in $N_d$ caused by anthropogenic aerosol emission due to the differing preindustrial (PI) and PD aerosol environments (Penner et al., 2011; Gryspeerdt et al., 2017), and (iv) the satellite sampling biases, which tends to discard clouds with high cloud fraction due to the inability of aerosol retrievals under cloudy conditions and thereby results in an artificial cloud regime selection (i.e., omitting more retrieval-reliable stratiform clouds; Gryspeerdt and Stier, 2012; Jia et al., 2021). Additionally, meteorological conditions, e.g., lower tropospheric stability (Ma et al., 2018a), relative humidity (Quaas et al., 2010), availability of water vapor (Qiu et al., 2017), and wind shear (Fan et al., 2009), and vertical overlapping status of aerosol and cloud layers (Costantino and Bréon, 2013; Zhao et al., 2019) also play roles in regulating aerosol-cloud interactions. It is worth noting that most of these studies calculated $S$ based on cloud effective radius rather than $N_d$, and so are subject to even more errors from the problem of stratification by liquid water path. Currently, a key difficulty in interpreting satellite observed aerosol–$N_d$ relationships is to isolate the causal impact of aerosols on $N_d$ from other confounding factors modifying the variations of aerosol and cloud simultaneously, specifically (i) updraft, determining cloud development as well as the maximum supersaturation at cloud base and thus aerosol population that can be activated, (ii) precipitation processes, depleting cloud droplets via coagulation and scavenging sub-cloud aerosol particles (iii) retrieval errors, biasing retrieved aerosol and cloud properties concurrently. However, a clear understanding on how they affect the estimates of $S$ quantitatively is lacking from the perspective of satellite observations (Quaas et al., 2020).

In terms of the updraft, in-situ aircraft measurements (Berg et al., 2011; Jia et al., 2019b), ground-based remote sensing (McComiskey et al., 2009; Schmidt et al., 2015), as well as detailed parcel model simulations (Reutter et al., 2009; Chen et al., 2016) clearly showed the dependency of $S$ on updraft, with generally larger $S$ at stronger updraft. In particular, co-variability of updrafts and aerosol concentrations has been found to result in a stronger $S$ than keeping vertical velocity ($w$) constant (Bougiatioti et al., 2020; Kacarab et al., 2020). As noted by Gryspeerdt et al. (2017), the updraft may roughly explain 20 % of the variability in $\Delta N_d$ from its PI-PD difference, adding to the uncertainty of the ERF$_{aci}$ estimate. Despite of the importance of dynamical constraint, it is not easily applicable to the analysis of satellite data due to the lack of updraft observation near cloud base at a global scale. As an alternative, cloud base height (CBH) may potentially serve as a practical proxy for the updraft at the base of liquid cloud because of their tightly linear correlation illustrated by in-situ observations of cumuliform clouds (Zheng and Rosenfeld, 2015). Although data used to draw this conclusion by Zheng and Rosenfeld (2015) were collected from only three locations, they covered various boundary conditions over both continent and ocean. Moreover, a theoretical framework has also been established to support the observed empirical relationship (Zheng, 2019), lending credibility to applying CBH as a proxy of the updraft. Building on this, recently developed CBH retrievals (Mülmenstädt et al., 2018; Böhm et al., 2019) offer an opportunity to gain some insight into the potential role updraft variability may play in the global ERF$_{aci}$ assessment.

In addition to the updraft, precipitation formation further complicates the derivation of the strength of $S$, since it can efficiently deplete cloud droplets and scavenge aerosols from clouds (Gryspeerdt et al., 2015). In such case, the change of $N_d$ is not necessarily related to actual aerosol perturbations (Chen et al., 2014) but rather to the intensity of cloud sink, thus in principle, should not be directly applied to infer $\Delta N_d$ driven by anthropogenic emissions. However, due to the lack of simultaneous observations of precipitation and aerosol/cloud properties from passive satellite remote sensing alone, most of ACI estimates did not consider the influence of precipitation (Quaas et al., 2008; Ma et al., 2014; Gryspeerdt et al., 2017; Jia et al., 2021) or just roughly identify the occurrence of rain relying on some simplified metrics, such as the threshold of 14 $\mu$m cloud effective radius (CER) for rain initiation (Gerber, 1996; Rosenfeld et al., 2019; Yang et al., 2021; Zhang et al., 2022) or the difference of CER between retrievals employing the bands of 2.1 and 3.7 $\mu$m (Saponaro et al., 2017; Jia et al., 2019a). Even though few studies have explicitly accounted for this by combining simultaneous precipitation observations from active remote sensing (Chen et al., 2014), how different treatments could influence the assessment of $S$ remains unclear. Solving this problem is helpful to reconcile the current diverse ACI estimates in order to achieve a more confident observational constraint.

For the satellite-based investigations, it is crucial but difficult to disentangle any physically meaningful attributable factors from artificial aerosol-cloud linkage induced by retrieval biases. In terms of $N_d$, retrievals for 3-D-shaped clouds and partially cloudy pixels deviate from the retrieval assumptions of overcast homogenous cloud and 1-D plane-parallel radiative transfer, thereby appear to lead to an overestimation of CER (Coakley et al., 2005; Matheson et al., 2006; Zhang and Platnick, 2011; Zhao et al., 2012), in turn, an underestimated $N_d$ (Grosvenor et al., 2018). This issue was reported to be more pronounced for broken cloud regimes, and could to some extent be addressed by only sampling $N_d$ for pixels with either high cloud fraction (Painemal et al., 2020) or large cloud optical depth (COT; Zhu et al., 2018). In addition to the assumptions on clouds, the existence of aerosols above clouds can also affect the retrieval of cloud optical depth (Haywood et al., 2004; Li et al., 2014), in turn bias $N_d$ calculation. Meanwhile, the retrieved AOD or aerosol index (AI) can be biased to a larger value due to the inability

to detect thin clouds in an aerosol-retrieval scene (Kaufman et al., 2005) or due to enhanced reflectance from neighbouring clouds (Várnai and Marshak, 2009). It is noteworthy that the overestimation of AOD tends to enhance with increasing cloud fraction (Zhang et al., 2005) and COT (Várnai and Marshak, 2021) as a result of both retrieval problems and aerosol swelling (Quaas et al., 2010). Therefore, the potential covariations between biases in $N_d$ and AOD (AI) modulated by cloud macrophysical properties could incur a spurious correlation between the two variables, obscuring the causal interpretation. While a few studies pointed out that the AOD(AI)-$N_d$ correlation is substantially enhanced when analyzing reliable $N_d$ retrievals (Jia et al., 2019a; Painemal et al., 2020), how and to which extent the satellite-diagnosed $S$ varies with the retrieval biases in terms of both aerosol and $N_d$, respectively, has not been fully understood. Such understanding is quite important for reconciling the previous estimates and proposing a meaningful method applicable to satellite-based investigations.

While the problem of vertical co-location between retrieved CCN proxy and clouds has been noticed in many previous studies, most of them placed focus on its influence on the correlation between aerosol and cloud (Stier, 2016; Painemal et al., 2020), i.e., a much higher correlation between $N_d$ and aerosol extinction coefficients near cloud base compared to $N_d$ vs. column-integrated aerosol quantity (AOD/AI), rather than the influence on $S$. The later is usually quantified as regression coefficient (regression slope in log-log space) between $N_d$ and CCN proxy and is a key determinant of radiative forcing estimates. Using AI as a CCN proxy, Costantino and Bréon (2010) demonstrated a weaker cloud susceptibility for the case with separated aerosol-cloud layers than well-mixed ones. However, it is unclear how the $S$ would change when switching commonly used column aerosol quantities to aerosol measures at cloud base. This understanding is particularly important for the inter-comparison and further reconciliation between current ACI metrics relying on diverse CCN proxies, including column-integrated, near-surface, and cloud level aerosol quantities.

In this study, we focus on the quantification of the impacts of three major confounders mentioned above, namely updraft, precipitation, and retrieval errors, as well as the problem of vertical co-location between aerosol and cloud, on the assessment of $S$ in the context of marine warm clouds by combining multiple active/passive satellite sensors and reanalysis products. On the basis on current findings, this study further suggests several potential ways forward to account for, to the extent possible, the major influencing factors practically for the satellite-based quantification of $S$, hence the $ERF_{aci}$.

## 2  Data and method

This work is based on observational data from multiple instruments on board Terra, Aqua and CloudSat platforms as well as reanalysis data from the Modern-Era Retrospective analysis for Research and Applications, version 2 (MERRA-2) (Randles et al., 2017) and the European Centre for Medium-Range Weather Forecasts (ECMWF) Reanalysis v5 (ERA5) (Hersbach et al., 2020). Table 1 summarizes the aerosol, cloud, and precipitation parameters and their corresponding sources, temporal-spatial resolutions, and time periods analyzed in the present study. Note that due to the requirement for co-located aerosol-cloud-precipitation observations, the data used in section 3.2 are obtained from the A-Train constellation of satellites (Aqua and CloudSat), which are then interpolated to 5×5 $km^2$ resolution for analysis, while the remaining parts are based on the observations from Terra, where all data are interpolated to $1° \times 1°$ resolution. The combination of datasets used in each section

is summarized in Table 2. It is worth mentioning that, as $S$ was found to vary with the spatial resolution of data (Sekiguchi et al., 2003; McComiskey and Feingold, 2012), the different data resolutions between section 3.2 and other sections can lead to a difference in $S$; but this is not the focus here. This study is restricted to global ocean with latitude between $60°$S and $60°$N because of limited quality of retrievals of aerosol size parameters (Levy et al., 2013) and $N_d$ (Gryspeerdt et al., 2021) over land and polar regions.

Aerosol properties (Levy et al., 2013) are obtained from the level 3 Moderate Resolution Imaging Spectroradiometer (MODIS) Dark Target product (MOD08 and MYD08; Platnick et al., 2017b). In order to collect co-located (adjacent) aerosol and cloud retrievals for analysis, aerosol retrievals on a coarse-resolved grid ($1° \times 1°$ on a latitude–longitude grid) are used to match cloud pixels ($1 \times 1$ km$^2$), assuming that aerosols properties in adjacent clear areas are homogeneous enough to represent those under cloudy conditions (Anderson et al., 2003; Quaas et al., 2008). Note that this assumption would be questionable

especially when aerosol is scavenged by precipitation (Gryspeerdt et al., 2015). In addition to commonly used AOD, aerosol index (AI = AOD × Ångström exponent) containing the information of aerosol size, is also employed since it is considered as a better proxy for CCN (Nakajima et al., 2001). The Ångström exponent is calculated from AOD at wavelengths of 460 and 660 nm. To eliminate $1°$ by $1°$ scenes where the aerosol distribution is heterogeneous, retrievals with a standard deviation higher than the mean values are discarded (Saponaro et al., 2017). As suggested by Hasekamp et al. (2019), the lowest 15 % of data

for AOD (AI) at a global scale are excluded to avoid large retrieval uncertainty at low aerosol concentrations (Ma et al., 2018a). Note that leaving out the low AOD (AI) yields a larger $S$ compared to using all data (Hasekamp et al., 2019).

    Cloud optical properties, including CER and COT at 3.7 $\mu$m (Platnick et al., 2017c), are obtained from the MODIS level 2 cloud products (MOD06 and MYD06; Platnick et al., 2017a), and then applied to compute $N_d$ based on the adiabatic approximation (Quaas et al., 2006). It was found that the filtering of cloud adiabaticity only has a negligible impact on the estimate of

$S$, but in turn results in a reduction of up to 63 % in the data volume (Gryspeerdt et al., 2021). For this reason, we do not apply such filtering here. Note that $N_d$ is calculated on the level of the satellite pixel (order 1 km) before aggregated to larger scales. Thus, the aggregation bias caused by the derivation of $N_d$ from the highly non-linear function of CER and COT as shown by Feingold et al. (2021), does not affect the results presented here. To ensure confident retrievals, the $N_d$ is filtered to include only single-layer liquid clouds with top temperature higher than 268 K. Pixels where CER < 4 $\mu$m and COT < 4 are discarded

due to the large uncertainty of retrievals (Sourdeval et al., 2016). In addition, only pixels with cloud fraction at 5 km resolution ($CF_{5x5km^2}$) > 0.9, and with a sub-pixel inhomogeneity index (cloud_mask_SPI) < 30 are used to reduce the retrieval errors induced by cloud edges and broken clouds (Zhang and Platnick, 2011). Further, we only consider pixels with a solar zenith angle of less than $65°$ and a sensor zenith angle of less than $41.4°$ to minimize the influence of known biases as detailed in Grosvenor et al. (2018). With the above sampling strategy, the random uncertainty in $N_d$ was reported at 78% on a pixel level

and this dropped substantially when averaged to a $1°$ by $1°$ region (Grosvenor et al., 2018). However, as stated in Gryspeerdt et al. (2021), the systematic bias in the $N_d$ retrievals to in situ measurements is low, with determination coefficients of 0.48 for all cloud types and 0.5-0.8 for stratocumulus clouds.

    To overcome the lack of the global updraft observation, we utilize satellite-based retrievals for CBH as a proxy of cloud base updraft for cumuliform clouds, based on the finding that these two quantities exhibit an approximately linear correlation

for convective clouds (Zheng and Rosenfeld, 2015). Here, clouds are considered convective for low troposphere static stability (LTS) less than 16 K (Rosenfeld et al., 2019). Additionally, cloud geometrical thickness (CGT; the difference between cloud top height and CBH) is used as an alternative proxy for the updraft regardless cloud types, since it has been observed to be associated with the cloud-base updraft for shallow cumuliform clouds (Lareau et al., 2018) and also correlated with cloud-base updraft for stratiform clouds via modulating cloud top cooling (Zheng et al., 2016). To obtain CBH and CGT, we apply a recently developed retrieval algorithm ($0.25° \times 0.25°$ resolution, Böhm et al., 2019) based on Multi-angle Imaging Spectro-Radiometer (MISR)/Terra observations, i.e. the MISR Level 2 Cloud Product (MIL2TCSP; NASA/LARC/SD/ASDC, 2012). The best performance of this algorithm is achieved for clouds with CBH around 1 km and CGT below 1 km. For such heights, which are characteristic for oceanic clouds considered in this analysis, the root mean square error ranges between 300–350 m. It is important to note that the MISR cloud-base height retrieval is limited to CBH > 560 m (Böhm et al., 2019). At this lower end of the detection range, a slight underestimation of the CBH is expected (Böhm et al., 2019). The ERA5 reanalysis is employed here to calculate LTS, as the difference in potential temperature between 700 and 1000 hPa (Klein and Hartmann, 1993). The hourly LTS is then matched to 10:30 local solar time to approximate the overpass time of the Terra satellite.

To identify the role of precipitation, CloudSat radar precipitation observations co-located with AOD/AI and $N_d$ from MODIS/Aqua are adopted as well. Here, we use the precipitation flag from the 2B-CLDCLASS product (Sassen and Wang, 2008) to distinguish precipitating (with the flags of 'liquid precipitation' and 'possible drizzle') and non-precipitation clouds (with the flag of 'no precipitation'). As a sink of $N_d$, drizzle could also affect the aerosol-cloud interactions even without rain falling on ground (Yang et al., 2021), so we also include drizzling clouds into precipitating cases. The CloudSat data at a $1.4 \times 2.5$ km$^2$ resolution are matched to the nearest MYD06 $5 \times 5$ km$^2$ pixels for further analyses.

The MERRA-2 product assimilates observations of the atmospheric state as well as remotely sensed AOD so that it can generate reasonable aerosol horizontal and vertical distributions (Buchard et al., 2017). The use of aerosol reanalysis also largely avoids the spuriously high AOD near clouds caused by the retrieval artifacts from satellite (Jia et al., 2021). Given that variability in sulfate aerosols contributes the most strongly to variability in $N_d$ among all aerosol species (McCoy et al., 2017), the sulfate concentration is considered be to the CCN proxy here. We utilize vertically resolved sulfate mass concentrations from MERRA-2 reanalysis in combination with the MISR CBH retrieval to obtain sulfate mass concentrations near cloud base (SO4B). In addition, sulfate surface mass concentrations (SO4S) and column mass density (SO4C) are also used to investigate if there will be different behaviors of $N_d$-to-CCN sensitivity when applying CCN proxies at different levels. The MERRA-2 3-hour averaged fields are interpolated to 10:30 local solar time to approximate the overpass time of the Terra satellite.

Figure 1 illustrates the regression procedure for calculating the $S$. After excluding the lowest 15 % AOD (AI), the data are then divided into 20 bins of CCN proxy, where each bin has an equal number of samples. The same number of samples ensures the same statistical representativeness within each bin. The values of $N_d$ and CCN proxy to a certain bin are the medians of all values in that bin. The generated 20 paired values of $N_d$ and CCN proxy are then used in linear regression to determine $S$ unless otherwise stated. The uncertainties of estimated $S$ is reflected by the 95 % confidence interval of the regression slope. We also tried 100 and 1000 bins, and found that the derived susceptibilities do not change significantly with number of bins. Additionally, the linear regression on all data points is also shown (yellow dashed line) in Fig. 1 for comparison with the

pre-binned approach, since both approaches have been used extensively by previous studies (Quaas et al., 2008; Gryspeerdt et al., 2017; Hasekamp et al., 2019; Rosenfeld et al., 2019) but it is unclear yet how large the difference in estimates between two approaches could be. Figure 1 illustrates that the pre-binned approach has a larger slope than lumping together all data points by 18 %, suggesting that attention should be paid when comparing $S$ derived from different approaches. In our study, both approaches lead to similar conclusions, as such, we will only focus on the results from pre-binned approach in the main text. Meanwhile, we also put the results associated with all-data approach to Supplementary Materials.

## 3   Results

### 3.1   Dependence on updraft

In adiabatic clouds, $N_d$ is essentially a function of both CCN and updraft (Feingold et al., 2001). To quantify how $N_d$ responds to CCN perturbations, the variation of updraft must be constrained. In a practical term, however, the observation of in-cloud vertical velocity is possible only from in-situ aircraft measurements or ground-based remote sensing, limiting the estimations to individual locations and sites. In order to obtain $S$ at a global scale, only possible from satellite, meteorological parameters (Ma et al., 2018b) or cloud regimes (Gryspeerdt and Stier, 2012) were generally employed to roughly approximate cloud dynamics. However, it should be noted that even in similar meteorological backgrounds and cloud regimes, the vertical velocity within individual clouds can still vary significantly (Hudson and Noble, 2014). Instead, based on previous findings from in-situ observations (see the section "Data and Methods"), our study utilizes CBH as a proxy of cloud base updraft for cumuliform clouds and CGT as a proxy for the updraft regardless cloud types. Note that with similar cloud top heights, the higher cloud base means thinner cloud layer. To avoid the potential interference by CGT, the analysis of the dependence of $S$ on CBH (Fig. 2a) is conducted within a quasi-constant CGT bin of 650–750 m. This range is chosen because of its relatively strong $S$, low possibility of precipitation as well as sufficient data points (Fig. 2b).

Figure 2 shows the dependence of linear regression slopes of ln $N_d$ versus ln AOD (ln AI), i.e., $S_{AOD}$ ($S_{AI}$), on CBH and CGT, respectively. To constrain the variation of cloud dynamics, the data are grouped over CBH and CGT bins with intervals of 80 and 100 m, respectively. It is seen that $S_{AOD}$ and $S_{AI}$ exhibit increases with both CBH and CGT, consistent with the expectation of stronger aerosol-cloud interactions under larger in-cloud vertical velocity conditions. The result is in accord with previous findings based on surface remote sensing under stratus (McComiskey et al., 2009) and altocumulus clouds (Schmidt et al., 2015). Also, using ground-based observations, Feingold et al. (2003) quantified this linkage and gave a correlation of 0.67 between $S$ and column maximum updraft. In our study, the correlation coefficients are 0.83 (0.98) for CBH–$S_{AOD}$ ($S_{AI}$) and 0.96 (0.95) for CGT–$S_{AOD}$ ($S_{AI}$). The higher correlations likely stem from the large volume of data used to stratify CBH(CGT), which enhances the representability of samples from a statistical perspective compared to the more limited number of cases used in Feingold et al. (2003).

It is also noted that, unlike the monotonic increase with CBH, $S_{AOD}$ ($S_{AI}$) increases with CGT at small-to-moderate CGT range (< 900 m) and then levels off (Fig. 2b). This is likely due to the tighter linkage between the occurrence of precipitation and CGT than CBH. Specifically, larger CGT is an indicator of strong updraft, tending to generate larger $S_{AOD}$ ($S_{AI}$), whereas

at the meantime it is also associated with the higher possibility of precipitation, which acts as an efficient sink of droplets (see section 3.2), thereby partly offsets the increase of $N_d$ induced by CCN, i.e., smaller $S_{AOD}$ ($S_{AI}$). In short, the situation of $S_{AOD}$

($S_{AI}$) at larger CGT (Fig. 2b) is a result of the competition between the effects of updraft and precipitation. Comparing the different CCN proxies, we see that in agreement with previous results (Hasekamp et al., 2019), $S_{AI}$ is consistently higher than $S_{AOD}$ for both all data cases (dashed lines) and almost all CBH (CGT) bins except for CGT > 900 m. For the remainder of the paper, only AI that is a better CCN proxy is used unless otherwise stated.

To gain insight into the mechanism underlying the apparent dependence of $S$ on updraft, we contrast AI–$N_d$ (CER) joint

histograms for weak and strong updraft conditions (Fig. 3). As the data volume for CBH case is too small to populate the joint histogram, only the CGT-related result is shown. Here, the subsets of data with CGT lower than the 25th percentile and higher than the 75th percentile are defined as weak and strong updrafts, respectively. Note that applying the 10th and 90th percentiles also yields similar results as shown in Fig. S2. It is known that the aerosol–$N_d$ relationship is nonlinear and, particularly, regime dependent. Reutter et al. (2009) proposed three distinct regimes according to the ratio of vertical velocity and aerosol

concentration: a) aerosol-limited regime, being characterized by high ratio value, nearly linear dependence of $N_d$ on aerosol, and insensitivity of $N_d$ to updraft, b) updraft-limited regime, being characterized by low ratio value and weak dependence of $N_d$ on aerosol but quite strong dependence on updraft, and c) transitional regime, falling between the above two regimes. Since we have limited the proxy of updraft (CGT) to a certain range, AI is thus assumed as an indicator of regime. Specifically, the low AI zone is more likely aerosol-limited while the high AI zone is close to updraft-limited regime. As illustrated in the

difference plots in Fig. 3, under the polluted condition with AI > 0.4, the samples of the strong updraft case tend to concentrate in the larger $N_d$ bins compared to the weak updraft (Fig. 3c), reflecting the critical role of updraft on facilitating activation of cloud droplets. Nevertheless, the distributions of CER do not exhibit systematic difference, except for less scattering for the strong updraft (Fig. 3f). As for the clean condition, what should be expected is the similar distribution of $N_d$ between different cloud dynamics as determined by the nature of aerosol-limited regime, or at least a slightly higher $N_d$ for the strong updraft

case. However, looking at the clean zone (AI < 0.15) in Fig. 3, it is clear that the strong updraft is associated with much lower $N_d$ as well as larger CER (generally larger than 14 μm, the threshold for drizzle initiation suggested by Freud and Rosenfeld (2012)) compared to the weak updraft, indicating a higher possibility of precipitation and/or drizzle. Consequently, the strong sink of droplets via precipitation at low AI and the enhanced activation of droplets at high AI will jointly create a much larger regression slope of ln $N_d$ versus ln AI for the strong updraft compared to the weak updraft condition. Moreover, these results

also imply that the interference of precipitation tends to amplify realistic dependence of $S_{AI}$ on the updraft, highlighting the need to remove the influence of precipitation on $N_d$ budget.

## 3.2 Dependence on precipitation

In this section, the role of precipitation on the quantification of $S$ will be explicitly accounted for by using the simultaneous aerosol-cloud-precipitation observations from CloudSat-MODIS combined datasets (see section 2). The hypothesis is that for

precipitating clouds, a sink to $N_d$ exists (via the coagulation) that is not reflecting the Twomey effect, so that the CCN - $N_d$ relationship is biased low in cases of precipitation formation. Figure 4 shows the AI-$N_d$ joint histograms for non-raining,

raining and all clouds as well as the difference between non-raining and raining cases. The raining clouds exhibit a lower $N_d$ relative to non-raining clouds over all AI bins, caused by the intensive sink of cloud droplets by collision–coalescence when precipitation forms (Fig. 4b,c,d). In addition, as the droplet sink and aerosol removal by precipitation can act together to veil the actual effect of aerosol on $N_d$, the $N_d$ in raining clouds shows a weaker response to increasing AI than that in non-raining clouds, with the corresponding $S_{AI}$ of 0.45 versus 0.56, respectively. The result is in agreement with Chen et al. (2014), who reported a consistently smaller CER-to-AI sensitivity in precipitating case than in non-precipitating case throughout different environmental conditions.

Interestingly, the regression slope of ln AI versus ln $N_d$ is enhanced after lumping all cloud scenes together regardless of whether it rains or not (Fig. 4a). The corresponding $S_{AI}$ (0.68) increases by 21 % relative to the non-raining case (0.56). This phenomenon was also noted by Painemal et al. (2020), and they speculated that drizzle appears to strengthen the aerosol–$N_d$ relationship, which is, however, contrary to the weaker $S_{AI}$ for raining clouds as illustrated above. For a clearer comparison of the $S_{AI}$ for non-raining, raining and all clouds, the fitting lines for these three cases are put into one single plot (Fig. 4e), with clean and polluted zones marked as blue and red, and the corresponding sample distributions are presented in Fig. 4f,g. It is shown that, the fitting line for all clouds nearly coincides with that for the non-raining case under polluted conditions, but closer to the raining case under clean conditions (Fig. 4e), consequently leading to a much steeper slope. This behavior is further corroborated by the different distributions of $N_d$. As shown in Fig 4g, the polluted clouds consist predominately of the non-raining clouds as a result of the suppression of precipitation by aerosols, thus maintaining a high value of $N_d$. Instead, the majority of the clean clouds are raining ones that are significantly subjected to the sink processes for $N_d$ and/or aerosol scavenging (Boucher and Quaas, 2013) (Fig. 4f), hence corresponding to a lower $N_d$. The results presented here imply that introducing the dependence of possibility of precipitation on aerosols (i.e., cloud lifetime effect) into the estimation of the Twomey effect, as commonly done in most previous studies, would perturb the statistical analysis and artificially bias the strength of the Twomey effect to a higher value. Moreover, it should be noted that a more extensive zone with $N_d$ being insensitive to aerosol is evident under low aerosol conditions after raining clouds being included (Fig. 4a), which means that, in addition to the overestimation of regression slope, the interference of precipitation also gives rise to an apparent non-linearity of the aerosol-$N_d$ relationship, hence adding substantial complexity in quantifying $S$ using a linear regression (Gryspeerdt et al., 2017).

## 3.3 Dependence on retrieval biases in AOD (AI) and $N_d$

Both aerosol retrievals errors due to 3D radiative effects and cloud contamination, aerosol swelling, and cloud retrieval errors for 3-D-shaped and heterogeneous clouds, have been shown to artificially introduce biases in the estimation of aerosol-cloud interactions (Quaas et al., 2010; Christensen et al., 2017; Neubauer et al., 2017; Jia et al., 2019a, 2021). Here, we dig deeper on how $S$ as a function of retrieval errors by defining two metrics that characterize the retrieval biases quantitatively. In order to obtain horizontally 'co-located' aerosol-cloud retrievals for analysis, the often adopted choice is a 1° by 1° gridding scale, at which aerosol concentrations are considered homogeneous (Anderson et al., 2003). Within a 1° by 1° grid box, sub-grid clear-sky and cloudy pixels co-exist (if clouds are not fully overcast) and are used for retrieving cloud and aerosol properties,

respectively. However, in case that most of clear-sky pixels are close to clouds, the problems of 3D radiative effects, cloud contamination, and aerosol swelling arise. Thus, the metric of aerosol retrieval errors (including 3D radiative effects and cloud contamination) and aerosol swelling is defined as the average distance to nearest cloudy pixel from clear pixels for aerosol retrieval ($\Delta L$), which is provided directly by MODIS L3 aerosol product. As for the cloud retrieval, the metric is the difference

between $N_d$ retrieved from all cloudy sub-pixels ($N_{dAll}$, without the cloud screening on CER, COT, $CF_{5x5km^2}$ and sub-pixel inhomogeneity index) and that retrieved from sub-pixels only with favorable situations for reliable cloud retrieval (see Methods for details), which is tightly related to the degree of cloud heterogeneity. Note that $N_{dAll}$ and $N_d$ are concurrently calculated for each $1°$ by $1°$ cloud scene, thus $\Delta N_d$ ($N_{dAll}$-$N_d$) only reflects the role of retrieval errors, with other conditions held constant (e.g., cloud types and meteorology). Generally, a negative value of $\Delta N_d$ is expected since a positive bias in CER and a negative

bias in COT for spatially inhomogeneous scenes act together to generate negatively biased $N_{dAll}$ according to the Equation 1 in Quaas et al. (2006). In this section, we also look at AOD in addition to AI, since AOD is a directly retrieved quantity and thus more closely related to retrieval problems.

Figure 5a shows the dependences of both AOD (AI) and linear regression slopes of ln $N_d$ ($N_{dAll}$) versus ln AOD (ln AI) on $\Delta L$. We note that AOD (AI) is the largest for the first $\Delta L$ bin with a value of 0.24 (0.17), and then drops rapidly to around 0.16

(0.13) for the other distances from clouds, indicating a quite strong near-cloud enhancement of AOD (AI) induced by retrieval biases and/or aerosol swelling. As AE was found to increase with $\Delta L$ (Várnai and Marshak, 2015), the reduction of AI with $\Delta L$ is thus less strong than AOD. Based on published in-situ aircraft measurements, we roughly isolate the contribution of aerosol swelling from retrieval issues (i.e., 3D radiative effects and cloud contamination). During the Indian Ocean Experiment (INDOEX), Twohy et al. (2009) measured a rise in relative humidity (RH) from about 70% at more than 20 km from cloud

to 90% with 1–4 km of cloud edge (equivalent to the distances of the third and first $\Delta L$ bins in Fig. 5a), which in turn results in about a 69% increase in aerosol scattering cross section (Twohy et al., 2009). Considering the aerosol humidification only occurs near cloud level, i.e., one quarter to one third of the aerosol column could be affected according to Lidar observations (Twohy et al., 2009), the increase in AOD by aerosol swelling is estimated to be 17–23%. This is up to about a third of relative increase in AOD from the third to first $\Delta L$ bins (64%) in Fig. 5a, implying that the retrieval errors in aerosol could contribute

the majority of the $S$ reduction in the first $\Delta L$ bin. It should be noted that the estimated AOD rise due to humidification relies on observed RH variability surrounding cloud and also the vertical profile and chemical composition of aerosol, which could vary with geographic location.

Corresponding to the biased high AOD (AI), $S_{AOD}$ and $S_{AI}$ for the first $\Delta L$ bin are greatly low relative to other bins, especially for AOD, suggesting that both retrieval biases and aerosol swelling near clouds could result in a severe underestimation in $S$.

These results imply that screening out the aerosol retrievals within the first $\Delta L$ bin (i.e., the average distance to the nearest cloud pixel less than 10 km) could be an applicable approach to sidestep the interference of aerosol retrieval biases. It is also noted that $S_{AOD}$ ($S_{AI}$) shows an increase first and then a decrease from the third $\Delta L$ bin. However, the following decrease is unlikely linked to the aerosol retrieval bias since the AOD (AI) remains almost constant (the upper panel in Fig. 5a). One interpretation for this would be that AOD/AI is getting less representative for the aerosol concentrations near cloud with increasing $\Delta L$, especially

for grid-boxes with precipitation where aerosol is not as homogeneous as assumed (Anderson et al., 2003). Moreover, as $\Delta L$

is also negatively correlated with CF (Várnai and Marshak, 2015), the decreasing $S_{AOD}$ ($S_{AI}$) is probably associated to other factors modulated by CF (such as retrieval error in $N_d$ as demonstrated in the following analysis).

Interestingly, Fig. 5a also depicts that the $S_{AOD}$ ($S_{AI}$) calculated from $N_{dAll}$ is consistently lower than that from $N_d$ for each $\Delta L$ bin, indicating that the cloud retrieval biases for partly cloudy pixels appear to lead to an underestimation of $S$. The increase of the difference between them with $\Delta L$ reveals that more underestimation occurs for high $\Delta L$ (typically low CF) conditions, where clouds are more partially cloudy, thereby deviate from the retrieval assumptions of overcast homogeneous cloud. As aforementioned, $\Delta N_d$ can act as a measure of some of the retrieval errors in cloud; the more negative $\Delta N_d$, the larger retrieval error in $N_d$. As shown in Fig. 5b, the $S_{AOD}$ ($S_{AI}$) calculated from $N_{dAll}$ increases with $\Delta N_d$, and then reaches its maximum when $\Delta N_d$ approaches 0, demonstrating that the satellite-diagnosed $S$ highly depends on the retrieval bias in cloud. In terms of the quality-assured $N_d$, the corresponding $S_{AOD}$ ($S_{AI}$) is not anticipated to be affected by retrieval issues, thus independent on $\Delta N_d$, but it is obviously not the case; the $S_{AOD}$ ($S_{AI}$) also significantly increases with $\Delta N_d$, which means that the criteria used for selecting homogeneous clouds within a 5 km × 5 km grid would not be as sufficient for an optimal performance of retrieval (Grosvenor et al., 2018) as we thought.

Figure 6 depicts relationships between $\Delta L$ and $\Delta N_d$, where the data are grouped as a function of CF for 50 cloud fraction bins containing same number of samples. It is clearly illustrated that CF regulates the negative correlation between $\Delta L$ and $\Delta N_d$. Under the condition of large CF, clear pixels are very close to the nearest cloud pixel, corresponding to a lower $\Delta L$, meanwhile, most of sub-grid cloud pixels meet the criteria for confident cloud retrievals, leading to a higher (near-zero) $\Delta N_d$; and the reverse is true in the case of low CF. This means that it is practically difficult to balance the accuracies of retrievals on both aerosol and cloud, since the aerosol retrieval should stay away from clouds, requiring low CF, whereas the $N_d$ retrieval should be performed in more homogeneous clouds (high CF) in order to satisfy the retrieval assumption of 1-D plane-parallel radiative transfer. To avoid the spuriously high AOD (AI) retrieval near clouds, the use of aerosol reanalysis would be a way forward (Jia et al., 2021). In terms of $N_d$, however, the situation is more complicated. Given that CF also correlates closely with cloud dynamics (CGT; Fig. 6), it does not make sense to simply restrict the analysis to low $\Delta N_d$ (thus high CF) to reduce the retrieval uncertainty of $N_d$; in doing so, a selection of cloud regime could be artificially applied.

## 3.4 Dependence on vertical co-location between aerosol and cloud

Currently, the use of reanalyzed/modeled aerosol vertical profiles seems the only feasible alternative to exploit the problem of vertical co-location since it is impossible yet to obtain aerosol retrievals below or within clouds from satellite (Stier, 2016; McCoy et al., 2017). Thus, unlike the previous sections based on satellite retrieved AOD/AI, vertically resolved SO4 from the MERRA-2 reanalysis is utilized here to obtain the CCN proxies for different altitudes. Although not as commonly adopted as AOD/AI, SO4C and SO4S were also used as CCN proxies by previous studies (McCoy et al., 2017; Jia et al., 2021). Here, the SO4C and SO4S are used, respectively, to mimic the behaviors of AOD/AI and surface aerosol extinction coefficient that are two commonly used CCN proxies in the satellite-based and ground-based methods (Quaas et al., 2008; Liu and Li, 2018), respectively. As demonstrated by Stier (2016), the SO4B derived in combination with CBH, is expected to be more relevant to the amount of CCNs actually activated at cloud base than SO4C and SO4S. The comparison of susceptibilities inferred from

these three proxies helps to understand whether the uses of column-integrated and near-surface aerosol quantities make sense, and more importantly, to reconcile the large range of existing estimates of the Twomey effect from different observational methods.

Figure 7 shows the two-dimensional probability density functions of $\ln N_d$ and $\ln$ SO4 along with fitting lines. We note that the pre-binned method yields similar high correlation coefficients ($R$) for SO4B (0.96), SO4S (0.95), and SO4C (0.98) due to the data stratification. When moving to the regression on all data points (Table S1), we can see that the $R$ for SO4B is the highest (0.6), followed by SO4S (0.57), and the $R$ for SO4C is the lowest (0.54), consistent with the results reported by (Stier, 2016) and (Painemal et al., 2020). In contrast, the regression slopes for SO4C (0.88) are, however, nearly twice as large as that for SO4B (0.47) and SO4S (0.46) (Fig. 7), implying that the strength of $S$ derived on a basis of column-integrated aerosol quantity, which is often the case for most previous satellite-based estimates, is overestimated by nearly a factor of two. Note that to explain the same change in $\ln N_d$, $\ln$ SO4B and $\ln$ SO4S increase by about 5, while $\ln$ SO4C only increases by 2 (Fig. 7). Translating to the linear scale, this means that SO4B (SO4S) increases by 148-fold, whereas only a tenfold increase can be seen in SO4C, resulting in the much larger slope of $\ln N_d$ versus $\ln$ SO4C. The underlying reason would be that the variability of SO4C is insufficient to explain the variabilities of SO4B (SO4S) .

In order to verify whether SO4C has the capability to capture the variability of SO4B quantitatively, the coefficient of variation (CV; calculated as the ratio of the standard deviation to the mean) is employed, which is a measure of relative variability, and particularly useful for the comparison among quantities with different magnitudes and units, e.g., SO4C (in units of $\mu g\ m^{-2}$) versus SO4B or SO4S (in units of $\mu g\ m^{-3}$) here. Since $S$ is generally inferred from the spatiotemporal variability of aerosol and cloud properties, here we calculate the temporal and spatial CVs, respectively; the temporal CV is calculated from the daily time series for the period 2006–2009 for each $1° \times 1°$ grid box, and the spatial CV is derived from the multi-annual averaged global geographical distribution. As shown in Fig. 8a,b,c, the temporal CVs of SO4C are smaller than those of SO4B and SO4S almost everywhere, with globally averaged CVs of 0.52 versus 1.02 and 1.03. Spatially, the larger CVs are generally located over the aerosol outflow regions, such as the western North Pacific, the Atlantic, and the east coasts of south America and South Africa, indicative of an impact of the strong variation of continental, and specifically anthropogenic emissions. Similarly, the spatial CV of SO4C exhibits a much smaller (0.88) value than those of SO4B and SO4S (1.84 and 1.79). In other words, the variability of SO4C is only able to reflect about half of the variability of SO4 near cloud base. This is mainly due to the important role of SO4 above cloud in total column SO4. However, above-cloud aerosol is much more homogeneous compared to SO4B and SO4S that are directly driven by rapid change of anthropogenic emissions near surface.

This is demonstrated in Fig. 8d, which shows that the ratio of SO4C below cloud (SO4BC) to SO4C is quite low, with a global average of 11.89 %. Spatially, the ratio can be up to 35 % over aerosol outflow regions, but generally below 10 % over vast remote oceans. The low ratio confirms the comparatively small sub-cloud aerosols in determining the aerosol loading within a column. Interestingly, there is also a good consistency between the spatial patterns of the ratio of SO4BC to SO4C and the correlation coefficient of SO4C with SO4B (Fig. 8d,e), i.e.,the high-ratio regions (the ratio > 15 %) generally have strong correlations ($R > 0.7$). Therefore, with regard to the vertical co-location, it is comparatively sensible to use column-

integrated quantities such as AOD/AI to represent CCN near cloud base over polluted continents and its immediate outflow region, where the correlation coefficient of SO4C with SO4B are overall larger than 0.7, but this is obviously not the case over remote oceans. The loose correlation between cloud-base and column-integrated aerosols found here ($R < 0.4$), in combination with the detectability limitations of satellite instrument on aerosol loading (Ma et al., 2018a), makes it more challenging to detect any meaningful aerosol-cloud associations in pristine environments from retrieved AOD/AI. Nevertheless, unlike the SO4C, rather strong correlations between SO4S and SO4B ($R > 0.7$) can be generally found with the only exception of high latitude oceans (Fig. 8f), which in combination with the highly similar aerosol-$N_d$ slopes and CVs between SO4S and SO4B, hints at surface observations as a promising way in terms of the vertical co-location issue.

## 4 Future improvements

Although this study has demonstrated the significant impacts of major confounders on the estimation of $N_d$-to-CCN sensitivity, some caveats remain. In order to achieve an optimal estimate of radiative forcing from the remote-sensing perspective, the following sources of uncertainty should be accounted for in future investigations.

The derivation of $N_d$ from satellite observations relies on a number of assumptions (Grosvenor et al., 2018), making it prone to systematic biases. While some sampling strategies have been applied to sidestep the biases in $N_d$ retrieval (see section 2), the uncertainties remain. To further ensure the cloud adiabaticity, there are two practical methods for use, including comparing the CER at different wavelengths (Bennartz and Rausch, 2017) and locating cloud "core" (Zhu et al., 2018). Appropriate $N_d$ sampling strategies are anyway beneficial in future investigations, though it has relatively little impact on $S$ (and the implied RF$_{aci}$)(Gryspeerdt et al., 2021).

The retrieved AOD (AI) as well as reanalyzed SO4 were treated as CCN proxies in this study. However, the usability is limited due to the lack of information on the aerosol size and/or hygroscopicity for AOD (AI), and also due to the fact that SO4 cannot fully explain the variability of CCN since organic aerosols also contribute significantly (Ruehl et al., 2016), particularly in the remote marine boundary layer (Zheng et al., 2020). Therefore, the application of direct CCN retrievals from polarimetric satellite (Hasekamp et al., 2019) is promising in future investigations of aerosol-cloud interactions. However, it would need to be combined with an estimate of the contribution of above-cloud aerosol especially in regions unaffected by continental outflow. More importantly, the PD CCN–$N_d$ relationship has been shown to be a better approximation of the PI and hence the "actual" sensitivity of $N_d$ to aerosol perturbations than AOD (AI)–$N_d$ relationship, as it is not affected by the differing PI and PD aerosol environments (Gryspeerdt et al., 2017). This highlights again the importance of directly retrieved CCN in the assessment of the radiative forcing from the Twomey effect.

Notably, using a linear regression slope from an ordinary least-squares (OLS) line fitting method to describe the actual nonlinear aerosol–$N_d$ (Fig. 1), can introduce additional uncertainties related to the problem of regression dilution (Pitkänen et al., 2016; Quaas et al., 2020). The OLS method is also likely to overestimate the change in $N_d$ from PI to PD over polluted continents, as a saturation effect will occur as aerosols keep rising under a polluted background. A joint-histograms method proposed by Gryspeerdt et al. (2017) can be useful to account for the nonlinearity.

In addition to the precipitation, entrainment mixing is a crucial droplet sink process (Blyth et al., 1988). However, given that it is practically difficult to infer a quantitative measure of the strength of entrainment mixing from satellite observations, its impacts were not considered explicitly here. It has been proven that entrainment mixing process is associated with dynamical and cloud regimes (Warner, 1969; de Roode and Wang, 2007), so the updraft-constraint in this study would also incorporate the effect of entrainment mixing to some extent. Although there have been some attempts to characterize entrainment mixing via the combination of lower tropospheric stability and relative humidity near cloud top (Chen et al., 2014; Jia et al., 2019a) or the $N_d$-LWP relationship at a certain phase relaxation time scale describing evaporation-entrainment feedback (Zhang et al., 2022), they are relatively rough approximations or qualitative differentiation. An updated approach for deriving measures of entrainment mixing at the global scale would be highly beneficial.

It was found that $S$ can vary not only with the spatial resolution of data (Sekiguchi et al., 2003; McComiskey and Feingold, 2012) but also with the spatial scale at which the regression is preformed (Grandey and Stier, 2010). Grandey and Stier (2010) demonstrated that conducting analysis over large regions could induce spurious aerosol-cloud correlations, mainly owing to the spatial co-variations in aerosol type, cloud regime, and meteorological conditions. Despite the global analyses employed in this study, the applied updraft constraint may make our results less susceptible to this issue. It is expected that, with joint use of updraft constraint and CCN retrieval that greatly eliminates the spatial gradient effects, the global analysis would be preferable compared to regional or local method, since the later could lead to a large bias in the aerosol–$N_d$ slope over pristine oceans where either the instrument detectability limitations on aerosol (Ma et al., 2018a) or the inability of column-integrated measure to represent aerosol near cloud base for low-aerosol condition (see section 3.4), could play a major role.

Given the impossibility to combine all datasets used in different sections together (e.g., the CBH/CGT from Terra are observed at 10:30 but the precipitation from Aqua at 13:30 local solar time), this work evaluates the individual impact of each bias on the estimate of $S$ separately. Nevertheless, the sources of bias could be also correlated with each other; thus an optimal estimate of $S$ with all biases constrained is desirable. Future studies are being planned to make use of CALIOP/CloudSat satellite observations, which provide simultaneous retrievals of aerosol extinction profiles, precipitation, and cloud base height (Mülmenstädt et al., 2018), such that an analysis accounting for all potential sources of bias can be performed.

## 5   Conclusions and discussions

By employing a statistically robust data set from multiple active/passive satellite sensors and reanalysis product, we systematically assessed the aerosol impact on marine warm clouds, and found that the $N_d$-to-CCN sensitivity ($S$) shows a strong dependence on a) updraft proxy, b) precipitation, c) satellite retrieval biases, as well as d) vertical co-location between aerosol and cloud layer. The key results and the corresponding implications are summarized as follows, and the impacts of issues highlighted here on the overall estimation of $S$ are listed in Table 3.

1. $S_{AOD}$ and $S_{AI}$ are found to increase remarkably with both CBH and CGT (treated as proxies for vertical velocity at cloud base), suggesting that stronger aerosol-cloud interactions generally occur under larger updraft velocity conditions. Although a similar dependency has been reported by some previous studies utilizing in situ aircraft measurements or ground-based

remote sensing, they were limited to certain time periods and regions. Instead, $S$ here is characterized as a function of CBH (CGT) based on 4 years of global satellite observations, which thus can reflect the full variability of cloud dynamic conditions. This functional relationship, as a better alternative of large scale meteorological conditions constraints (less directly linked to cloud dynamics in a cloud scale) could be promising in application to the estimation of global aerosol-cloud radiative forcing,

by which the change in $N_d$ from the PI to the PD may be inferred based on CBH (CGT) climatology from satellite and anthropogenic aerosol emission perturbation assuming to first order un-changed CBH distributions.

     2. There exists an intensive sink of cloud droplets by precipitation, thereby leading to a much lower $N_d$ in raining clouds (55 cm$^{-3}$) compared to non-raining clouds (125 cm$^{-3}$). In turn, a weaker $S$ was found in raining clouds than that in non-raining clouds, with the corresponding $S_{AI}$ of 0.45 versus 0.56, respectively. Surprisingly, after lumping all cloud scenes together,

the derived $S_{AI}$ (0.68) is amplified by 21 % (51 %) relative to the non-raining (raining) case, and also a more non-linear aerosol-$N_d$ relationship is diagnosed. We showed that this amplification is just an artifact governed by the joint impacts of the suppression of precipitation by aerosols and the aerosol removal by precipitation. That is, introducing the confounding effect of aerosol-precipitation interactions into the estimation of the Twomey effect can artificially bias the $S$ to a higher value. The finding highlights the necessity of removing precipitating clouds from statistical analyses when quantifying $S$ and assessing

the Twomey effect. To achieve this, the only way would be simultaneous aerosol-cloud-precipitation retrievals (e.g., from the A-Train satellite constellation). However, due to the fact that most of existing estimates of $S$ and its radiative forcing did not take this aspect into consideration, the relative change of $S_{AI}$ from the all clouds to non-raining clouds presented here could serve as a useful reference for the inter-comparison of the cloud susceptibilities from different studies.

     3. Aerosol retrieval biases (3D radiative effects and cloud contamination), aerosol swelling, and cloud retrieval bias (het-

erogeneity effect) tend to lead to an underestimation of $S$. Although $S_{AI}$ ($S_{AOD}$) for the first $\Delta L$ bin, where evident AI(AOD) enhancement exists, is about 29% (50%) less than other unaffected bins, the overall underestimation is only ∼3% because of the small data volume in the first bin (Fig. 5a). Nevertheless, for low-$\Delta L$ dominated regions (e.g., stratcumulus regions), the underestimation can be more pronounced. By comparing $S_{AI}$ ($S_{AOD}$) calculated by $N_{dAll}$ and $N_d$, the underestimation by cloud retrieval issues is roughly estimated to be ∼8% (∼17%). It is noted that the CF can act as a key modulator of these two

kinds of retrieval issues, i.e., an increase in CF enhances the aerosol retrieval biases via intensifying near-cloud enhancement of AOD (AI) but reduces cloud retrieval errors via alleviating the cloud heterogeneity, making it practically difficult to balance the accuracies of both retrievals within a same grid. In terms of aerosol, the use of aerosol reanalysis is a potential way to avoid the near-cloud enhancement of AOD (AI), but note that the issue of aerosol swelling remains to some extent. As for $N_d$, the retrievals under high CF (over a 1°×1° grid) condition would be preferable even though strict criteria for cloud screening

(Grosvenor et al., 2018) have been applied, which, however, could incur an artificial selection of cloud regime since CF also covaries with cloud dynamics. Therefore, applying a CF-updraft constraint in the $N_d$ screening would be a path forward.

     4. Use of vertically integrated SO4 (SO4C) as a proxy of CCN near cloud base results in a degradation of correlation with $N_d$, with an approximately two-fold enhancement of $S$ as compared to using SO4 near cloud base (SO4B). This is mostly attributed to the inability of SO4C to capture the full variability of SO4B. Generally, SO4C is dominated by SO4 above cloud,

which is relatively homogeneous compared to SO4B that is tightly linked to rapid changes of anthropogenic but also natural

emissions near surface. As a result, to explain the same change of $N_d$, the corresponding fractional change in SO4C is much smaller than SO4B, hence leading to a higher regression slope that, however, is not associated with physically meaningful enhancement of $S$. The similar aerosol-$N_d$ slopes, correlation coefficients as well as relative variability between SO4S (SO4 near surface) and SO4B, suggest that the use of near-surface aerosol measurements, such as particulate matter (Guo et al., 2018) or aerosol extinction coefficients (Liu and Li, 2018), is an effective solution to the problem of vertical co-location in the case that observations of vertical profile of aerosol and cloud base height are unavailable, although its suitability would depend on the degree of coupling of boundary layer (Painemal et al., 2020). Moreover, the result further raises complications to compare and reconcile the diverse cloud susceptibilities from studies utilizing CCN proxies at different altitudes. It should be noted that the derivation of $N_d$ change from PI to PD (thus radiative forcing) is expected to be less affected, given that the vertical co-location issue also applies to fractional change of aerosol due to anthropogenic emissions, thus partly compensating the enhancement of $S$; nevertheless, the net effect on radiative forcing still needs further exploration.

*Data availability.* The MODIS Aqua and Terra Level 3 products are available from https://doi.org/10.5067/MODIS/MYD08_D3.061 and https://doi.org/10.5067/MODIS/MOD08_D3.061, and Level 2 products are available from https://doi.org/10.5067/MODIS/MYD06_L2.061 and https://doi.org/10.5067/MODIS/MOD06_L2.061. The CloudSat data is available from http://cloudsat.atmos.colostate.edu/data/. The MISR Level 2 Cloud Product (MIL2TCSP) data are from https://asdc.larc.nasa.gov/data/MISR/MIL2TCSP.001/. The MERRA-2 and ERA5 reanalysis products are collected from https://goldsmr4.gesdisc.eosdis.nasa.gov/data/MERRA2/ and https://www.ecmwf.int/en/forecasts/dataset/ecmwf-reanalysis-v5.

*Author contributions.* HJ and JQ designed the research. HJ performed the research and prepared the manuscript, with comments from JQ, EG, CB, and OS.

*Competing interests.* JQ and OS are Associate Editors of ACP. The authors declare that they have no conflict of interest.

*Acknowledgements.* MODIS data were acquired from the Level-1 and Atmosphere Archive & Distribution System (LAADS) Distributed Active Archive Center (DAAC). MISR products were obtained from the NASA Langley Research Center Atmospheric Science Data Center. CloudSat data products were provided by the CloudSat Data Processing Center at the Cooperative Institute for Research in the Atmosphere, Colorado State University. MERRA-2 reanalysis products were provided by NASA's Global Monitoring and Assimilation Office (GMAO). ERA5 reanalysis data sets were retrieved from ECMWF's Meteorological Archival and Retrieval System (MARS). This work also used JASMIN, the UK's collaborative data analysis environment (http://jasmin.ac.uk). The authors gratefully acknowledge funding by the German Research Foundation (Joint call between National Science Foundation of China and Deutsche Forschungsgemeinschaft, DFG,

GZ QU 311/28-1, project "CloudTrend") and by the European Union Horizon2020 project FORCES (grant agreement no 821205). EG was supported by a Royal Society University Research Fellowship (URF/R1/191602).

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

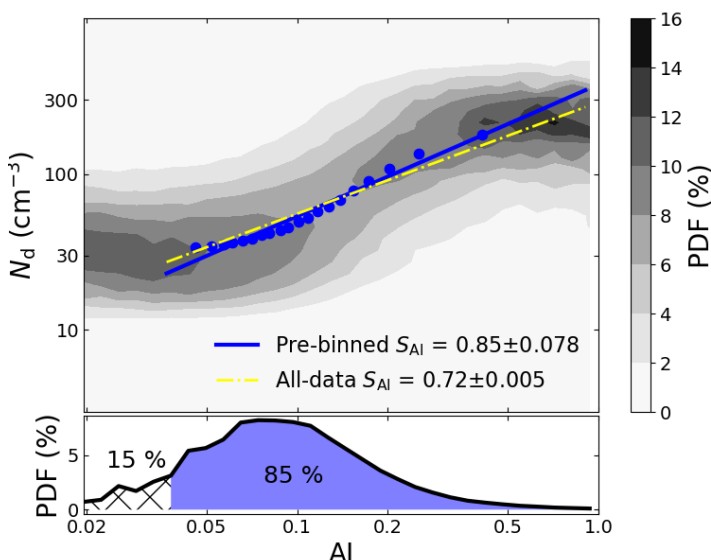

**Figure 1.** Schematic diagram of the procedure for calculating the sensitivity (linear regression coefficient in log–log space) of $N_d$-to-CCN, where AI is taken as a example. Upper panel shows the global joint $N_d$–AI histogram, where each column is normalized so that it sums to 1. The blue line is a linear regression on the 20 paired $N_d$-AI (blue dots) that are the medians of each AI bin with an equal number of samples, and the yellow dashed line shows a linear regression on all data points. Note that the lowest 15 % AI have been left out according to its occurrence (bottom) before binning data. The cloud susceptibilities to AI ($S_{AI}$) derived from both approaches are shown along with 95 % uncertainty estimates (according to Student's t-test). Data used here are the same as in section 3.1.

.

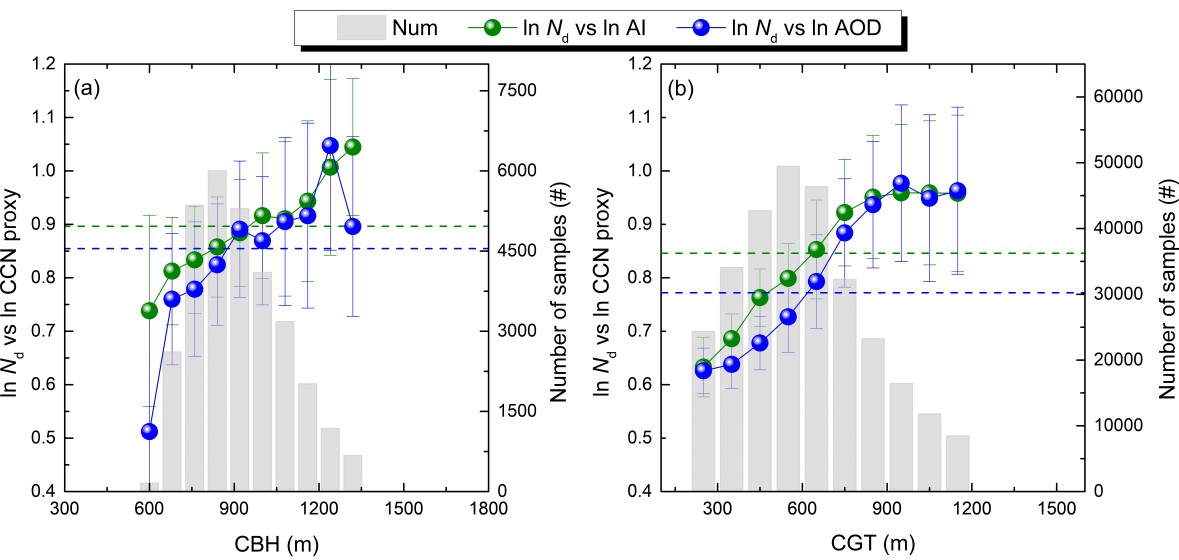

**Figure 2.** Dependence of the linear regression slopes of ln $N_d$ versus ln AOD (blue) and ln AI (green) on (a) CBH and (b) CGT derived via the pre-binned approach. Data are grouped into 10 fixed CBH (CGT) intervals for regressions. Error bars indicate the 95 % confidence interval of the linear regression, and the gray bars denote the total number of samples for each CBH (CGT) bin. The corresponding regression slopes computed from the data over all CBH (CGT) bins are shown as horizontal dashed lines (green for AI and blue for AOD). The equivalent Fig. S1 shows similar results based on the all-data approach.

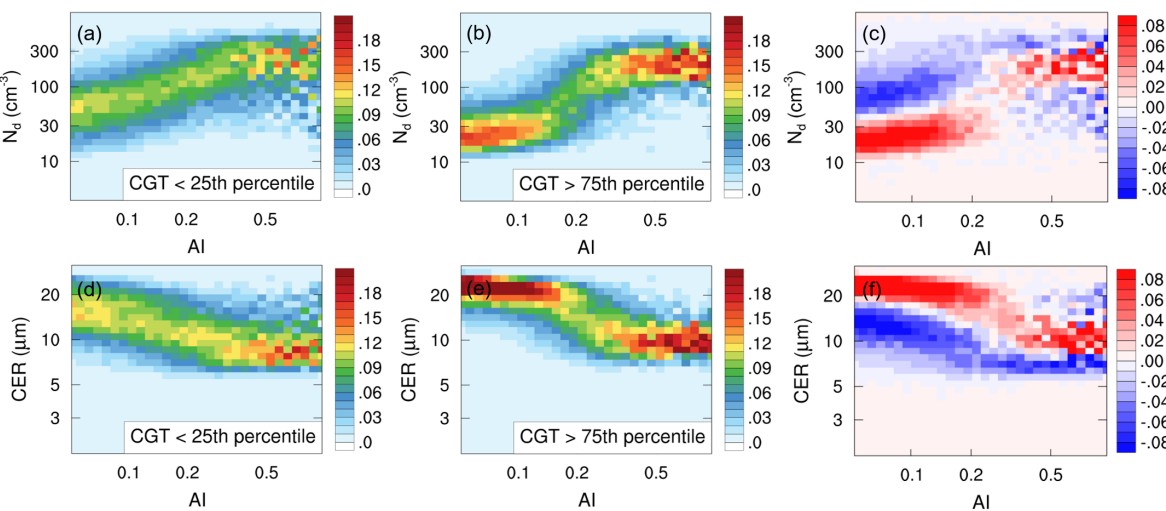

**Figure 3.** Joint histograms between AI and $N_d$ (CER) created for weak and strong updraft conditions, as defined by the lowest and the highest CGT quartiles, respectively. The difference plots between strong and weak cases are shown at the end of each row. The histograms are normalized so each column sums to 1, such that the histograms show the probability of observing a specific $N_d$ (CER), given a certain AI.

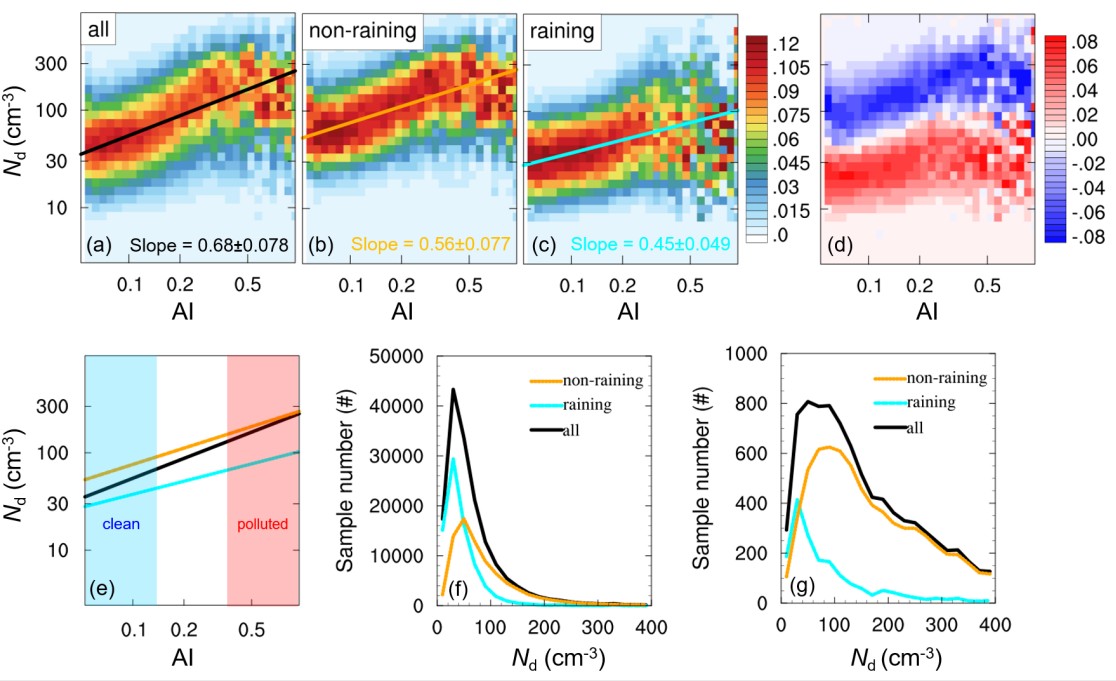

**Figure 4.** Joint histograms between AI and $N_d$ created for (a) all clouds, (b) non-raining, and (c) raining clouds, as well as (d) the difference of joint histograms between the raining and non-raining cases. Cloud susceptibilities to AI derived via the pre-binned approach are also shown along with 95 % uncertainty estimates (according to Student's t-test). The fitting lines for three cases are merged into one single plot (e), with clean and polluted zones marked as blue and red, and the corresponding sample distributions are also shown (f, g).

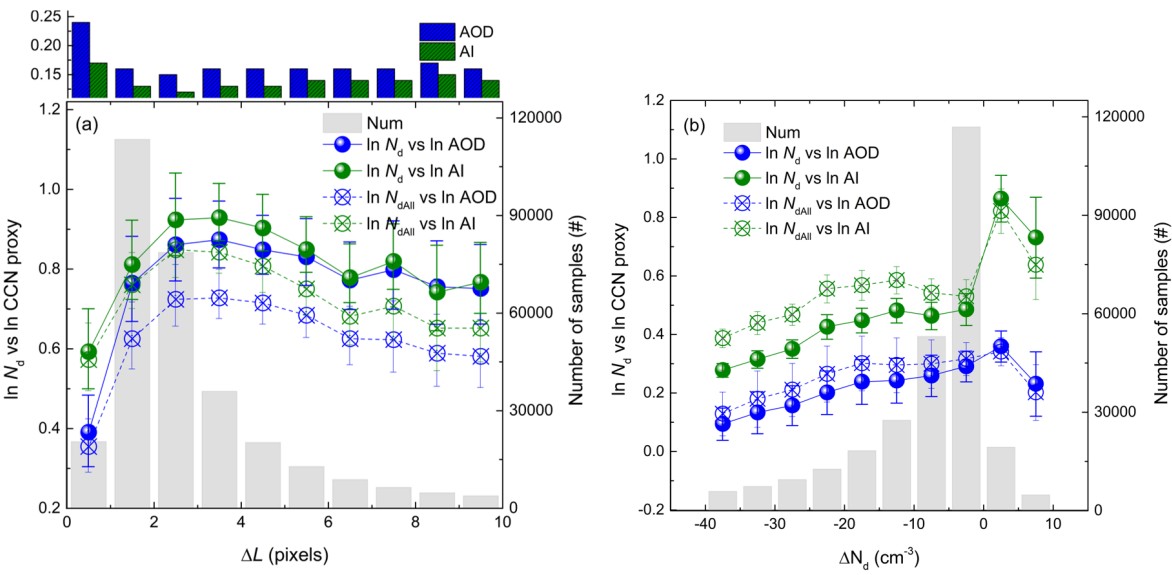

**Figure 5.** Dependence of the linear regression slopes of $\ln N_d$ ($\ln N_{dAll}$) versus $\ln$ AOD ($\ln$ AI) on (a) $\Delta L$ and (b) $\Delta N_d$ derived via the pre-binned approach. Data are grouped into 10 fixed $\Delta L$ ($\Delta N_d$) intervals for the calculation of slopes. Error bars indicate the 95 % confidence interval of the linear regression, and the gray bars denote the total number of samples for each bin. The change of AOD (AI) with $\Delta L$ is also shown in the panel (a). The equivalent Fig. S3 shows similar results based on the all-data approach.

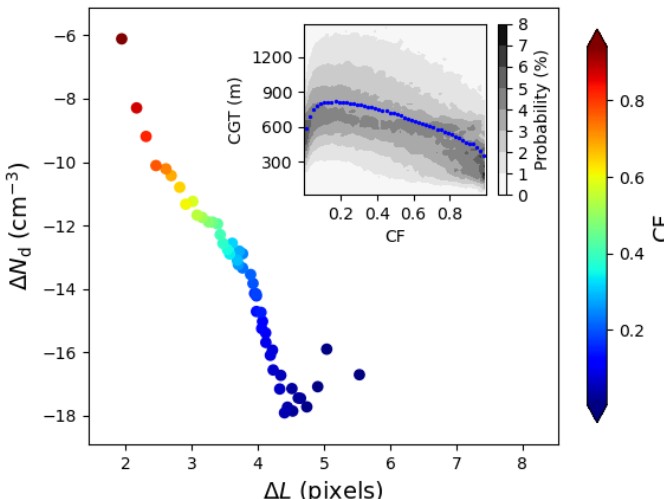

**Figure 6.** Relationships between $\Delta L$ and $\Delta N_d$, where the data are grouped as a function of CF with each CF bin containing same number of samples. Joint histogram between CF and CGT is shown in the inner plot, where the blue dot shows the median CGT at each CF bin.

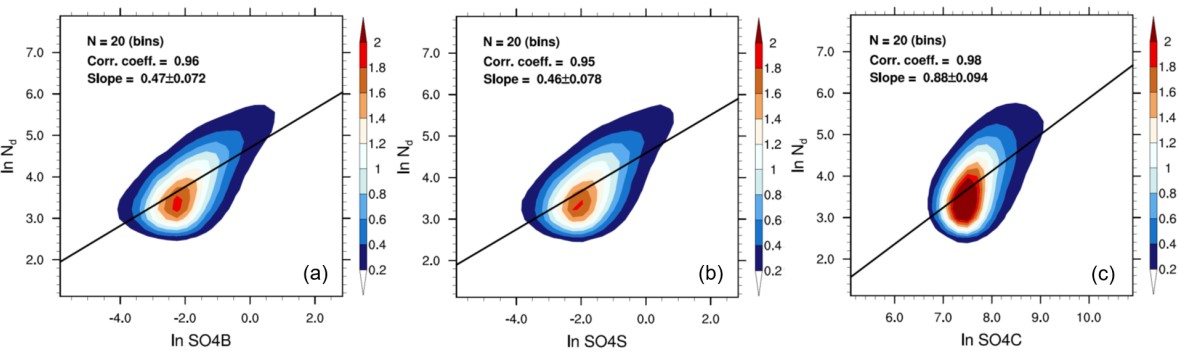

**Figure 7.** Two-dimensional probability density functions of ln $N_d$ versus (a) ln SO4B, (b) ln SO4S, and (c) ln SO4C, respectively, for the period 2006-2009. Sample numbers ($N$), correlation coefficients, and regression slopes with 95 % uncertainty estimates (according to Student's t-test) for pre-binned SO4-$N_d$ pairs are displayed in the upper left of each plot.

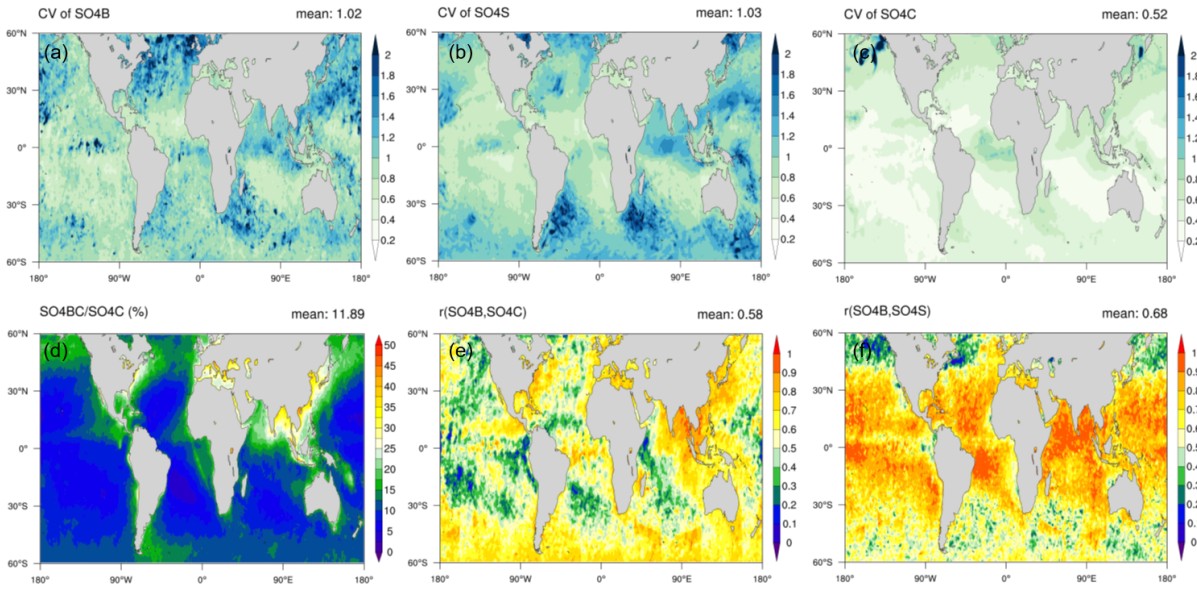

**Figure 8.** Map of coefficients of variations (CV) of (a) SO4B, (b) SO4S, and (c) SO4C, (d) ratio of column mass of SO4 below clouds (SO4BC) to SO4C (%), Pearson's correlation coefficients of SO4B with (e) SO4C and (f) SO4S, which are calculated for each 1°×1° grid box over the period 2006–2009.

**Table 1.** The list of the parameters, sources, and their corresponding temporal-spatial resolutions applied in present study.

| Source | Time period | Resolution | Parameters |
|---|---|---|---|
| MYD08/MOD08 | Jan 2008–Dec 2008 for MYD08<br>Jan 2006–Dec 2009 for MOD08 | Daily, 1°× 1° | AOD at 460/550/660 nm[a]<br>Distance to nearest cloudy pixel ($\Delta L$)<br>CF |
| MYD06/MOD06 | Jan 2008–Dec 2008 for MYD06<br>Jan 2006–Dec 2009 for MOD06 | Daily, $1 \times 1$ km$^2$<br><br><br><br><br><br>Daily, $5 \times 5$ km$^2$ | COT at 3.7 μm<br>CER at 3.7 μm<br>Cloud_Mask_SPI<br>Cloud top temperature<br>Cloud multi-layer flag<br>Cloud phase flag<br>$CF_{5x5km^2}$<br>Solar zenith angle<br>Sensor zenith angle |
| CloudSat | Jan 2008–Dec 2008 | Daily, $1.4 \times 2.5$ km$^2$ | Precipitation flag |
| MISR | Jan 2006–Dec 2009 | Daily, 0.25° × 0.25° | CBH<br>CTH |
| MERRA-2 | Jan 2006–Dec 2009 | 3-hourly, 0.5°×0.625° | Sulfate mass mixing ratio profile<br>Air density |
| ERA5 | Jan 2006–Dec 2009 | hourly, 0.25°×0.25° | Temperatures at 700 and 1000 hPa |

**Table 2.** The combination of datasets used in each subsection of the Results section.

| Subsection | Datasets |
|---|---|
| Section 3.1 | MOD08, MOD06, MISR, ERA5 |
| Section 3.2 | MYD08, MYD06, CloudSat |
| Section 3.3 | MOD08, MOD06, MISR |
| Section 3.4 | MERRA-2, MOD06, MISR |

**Table 3.** Issues highlighted in this study and their impacts on the overall estimation of *S*.

| Process not considered | Impact on *S* |
|---|---|
| Updraft dependency | To be assessed |
| Precipitation | Biased high by ∼21% (∼29%) for AI (AOD) |
| Aerosol retrieval bias and aerosol swelling | Biased low by ∼3%(∼3%) for AI (AOD) |
| Cloud retrieval bias | Biased low by ∼8% (∼17%) for AI (AOD) |
| Vertical co-location between aerosol and cloud | Biased high by ∼87% |