# Peer review of "Addressing the difficulties in quantifying droplet number response to aerosol from satellite observations"

_Atmospheric Chemistry and Physics, 2021_

## Author Comment (AC1)

**Response to the Reviewer's Comments**

March 15, 2022

We would like to thank the reviewer for the effort in helping us improve the manuscript. Below we respond point-by-point to the comments, with the reviewer comments in black, and our responses in blue. The manuscript has been revised accordingly. The line numbers in the response are for the revised version with tracked changes.

**Comments by Reviewer #1**

This study investigates the changes of Twomey effect for marine warm clouds with various influential factors including the updraft, precipitation, retrieval errors, and vertial co-location between aerosol and clouds. Valuable results have been obtained, which can improve our understanding of the radiative impacts from aerosol-cloud interaction from perspective of satellite observations. Also, the paper is well written. I personally think this manuscript is suitable for publication after a minor revision.

We thank the reviewer for this thoughtful and constructive statement. We have revised the manuscript carefully according to the reviewer's comments. Please see the following detailed point-by-point responses.

**Detail comments**

1. Line 19-21, In addition to the radiative impacts of aerosols by serving as CCN, aerosols can also affect the development of clouds and then precipitation and radiation by modifying the thermal structure of atmosphere via direct radiative effect.

   Thanks for the comment. This effect is now included in the introduction on lines 23-24.

2. Line 24-25, regarding the rapid adjustments, one reference is suggested here which showed the increase of cloud liquid water path and decrease of cloud re with incrased Nd via Twomey effect, Zhao and Garrett (2015, doi: 10.1002/2014GL062015).

   Thanks for bringing this study to our attention. We now cite this paper in the revised text (line 29).

3. Line 28-29, Actually even with the same climate model simulation (such as CAM5), the aerosol first indirect effect also varies with the aerosol variables that are used to present the aerosol amount.

   We agree with the reviewer that the choice of aerosol proxy can play a critical role. This point was discussed in the original manuscript (now lines 41-42).

4. Line 33-41, There are various influential factors, which are not limited to these five points. For example, the existence of precipitation particles within clouds as indicated by Yang et al. (2021, doi: 10.1029/2021JD035609) based on satellite observations, the aerosol amount or availability of water vapor amount as indicated by Qiu et al. (2017, doi: 10.1016/j.atmosenv.2017.06.002), cloud types or vertical locations as indicated by Zhao et al. (2019, doi: 10.3390/atmos10010019), and potential large uncertainties in cloud retrievals as indicated by Zhao et al. (2012, doi: 10.1029/2011JD016792), and so on.

   Thanks for the comment. These influential factors have been discussed and the relevant references have been cited in the introduction.

5. Line 54-56, Good idea. However ,with this assumption or method, we may limit the cloud types as cumuliform clouds.

   We thank the reviewer for this very important comment. To ensure the applicability of CBH as a updraft proxy. we now restrict the analyses related to CBH to convective clouds only by adopting the threshold of LTS<16 K (Rosenfeld et al., 2019). A similar dependence, i.e., increasing S with CBH, is also seen even after

constraining the cloud type (Figure 2a in the revised manuscript). It is worth mentioning that, beyond the CBH, the CGT was used as an alternative proxy for the updraft regardless the cloud types, since it has been observed to be associated with the cloud-base updraft for shallow cumuliform clouds(Lareau et al., 2018) and also correlated with cloud-base updraft for stratiform clouds via modulating cloud top cooling (Zheng et al., 2016). Therefore, the conclusion of updraft-dependency holds for all cloud types. We now clarify these in the method section on lines 160-166, and the Figure 2a is revised accordingly.

6. Line 65-66, The reference mentioned above (Yang et al. 2021) also took use of the simple threshold value method with 14 um.

   Thank you. The paper is now cited in the revised manuscript.

7. Line 69, Do the authors mean "Solving this problem is helpful to ..."?

   Thanks for the comment. We have rephrased the sentence as suggested by the reviewer (line 85).

8. Line 77-81, Actually, the existence of aerosols could also cause biases to satellite-based cloud retrievals. As indicated by Li et al. (2014, doi: 10.1002/2013JD021053), the existence of absorbing aerosols could cause the satellite based retrieval of optical depth lower, effective radius higher, and so on.

   Thanks for the comment. We now cite this paper and discuss the retrieval uncertainty on lines 94-95.

9. Line 107-110, previous studies have already indicated that the aerosol-cloud interaction is sensitive to the spatial resolution. How do the authors consider this point?

   Thanks for this comment. The reviewer is right that the varying spatial resolution can also play a role. We did not explicitly consider this issue since this is not the focus of this study, but a discussion on this issue is now added on lines 130-133.

10. Line 110-111, It is well known that the retrieval uncertainties are large over polar regions, how about that over land regions? A reference might be helpful.

    Thanks for the suggestion. We do this in the revised manuscript (line 134).

11. Line 112-125, why are the Level 3 aerosol data but Level 2 cloud data used in this study?

    This is because the use of L2 cloud data can avoid the aggregation bias of $N_d$ calculation (Feingold et al., 2021), and also enable us to select more reliable cloud retrievals on the level of the satellite pixel (1x1 km2) with different flags as detained on lines 154-159. These reliable cloud retrievals are then aggregated to larger scales to match Level 3 aerosol retrievals.

12. Line 123, it might be better used as Feingold et al. (2021)

    Thanks for the reminder. Corrected on line 153.

13. Line 130-136, what are the potential limiations or uncertianties in the cloud base height retrievals by the introduced method? It is worthy to briefly describe.

    The best performance of this algorithm is achieved for clouds with CBH around 1 km and CGT below 1 km. For such heights, which are characteristic for oceanic clouds considered in this analysis, the root mean square error ranges between 300–350 m. It is important to note that the MISR cloud-base height retrieval is limited to CBH > 560 m (Böhm et al., 2019). At this lower end of the detection range, a slight underestimation of the CBH is expected (Böhm et al., 2019). This is now clarified on lines 170-174.

14. Line 136-141, even within non-precipitating clouds, drizzle could exist and affect the aerosol-cloud interaction, as indicated by the reference mentioned earlier Yang et al. (2021), how could the authors consider this impact?

    Thanks for the question. As the drizzle affects the aerosol-cloud interactions in a same manner with rainfall, i.e., as a strong sink of $N_d$, we considered both 'liquid precipitation' and 'possible drizzle' as precipitating clouds (see line 179). The important result of Yang et al. (2021) is now mentioned on lines 180-181.

15. Line 180-181, One possibility is the large volume of datasets. Could the data selection also play a role to the higher correlation?

    The reviewer has a great point. It might somehow play a role. However, given the similar cloud type, i.e., ice-free and single layered clouds, selected in both studies, we can not conclude the data selection is the reason for the higher correlation.

16. Line 200-201, it is easy to understand that the low AI zone is more likely aerosol-limited. However, I cannot understand why the high AI zone is close to updraft-limited regime if we do not know how large the updraft is? Could the authors expalin more?

Thanks for the question. As we have limited the proxy of updraft to a certain range ($<$ the 10th/25th percentile or $>$ the 75th/90th percentile) in Fig.3 and Fig. S2, it is thus assumed that the ratio of updraft to AI is dominated by AI. A statement on this is now added on line 248

17. Line 226, I would suggest using the same format, either with or without parathesis for ln AI.

   Thanks for the reminder. We now remove the parentheses of ln (AI), ln ($N_{\mathrm{d}}$), ln(SO4C/B/S) throughout the text.

18. Line 233-237, if possible, I personally would like to suggest seperating this long sentence to a few short sentences.

   Thank you for the suggestion. This sentence is now split into three short ones (lines 288-290).

19. Line 270, 'appears to'-> 'appear to'

   Thank you. Corrected.

20. Line 317, why do the authors choose to use daily time series values instead of hourly?

   Because aerosol and cloud properties are linked on a daily basis throughout the manuscript, we thus choose daily time series to derive the temporal CV so as to quantify the day-to-day variability of SO4.

**References**

Böhm, C., Sourdeval, O., Mülmenstädt, J., Quaas, J., and Crewell, S.: Cloud base height retrieval from multi-angle satellite data, Atmos. Meas. Tech., 12, 1841–1860, https://doi.org/10.5194/amt-12-1841-2019, 2019.

Feingold, G., Goren, T., and Yamaguchi, T.: Quantifying Albedo Susceptibility Biases in Shallow Clouds, Atmos. Chem. Phys. Discuss., 2021, 1–31, https://doi.org/10.5194/acp-2021-859, 2021.

Lareau, N. P., Zhang, Y., and Klein, S. A.: Observed Boundary Layer Controls on Shallow Cumulus at the ARM Southern Great Plains Site, J. Atmos. Sci., 75, 2235–2255, https://doi.org/10.1175/JAS-D-17-0244.1, 2018.

Rosenfeld, D., Zhu, Y., Wang, M., Zheng, Y., Goren, T., and Yu, S.: Aerosol-driven droplet concentrations dominate coverage and water of oceanic low-level clouds, Science (80-. )., 363, https://doi.org/10.1126/science.aav0566, 2019.

Yang, Y., Zhao, C., Wang, Y., Zhao, X., Sun, W., Yang, J., Ma, Z., and Fan, H.: Multi-Source Data Based Investigation of Aerosol-Cloud Interaction Over the North China Plain and North of the Yangtze Plain, J. Geophys. Res. Atmos., 126, https://doi.org/10.1029/2021JD035609, 2021.

Zheng, Y., Rosenfeld, D., and Li, Z.: Quantifying cloud base updraft speeds of marine stratocumulus from cloud top radiative cooling, Geophys. Res. Lett., 43, https://doi.org/10.1002/2016GL071185, 2016.

---

## Author Comment (AC2)

**Response to the Reviewer's Comments**

March 15, 2022

We would like to thank the reviewer for the effort in helping us improve the manuscript. Below we respond point-by-point to the comments, with the reviewer comments in black, and our responses in blue. The manuscript has been revised accordingly. The line numbers in the response are for the revised version with tracked changes.

**Comments by Reviewer #2**

The article studies the effect of aerosols on liquid cloud droplet concentration using several space-based instruments (active and passive) to describe cloud and aerosol properties and reanalysis to retrieve information on sulfate. The authors investigate the potential biases that are usually overlooked by most of the studies quantifying this effect. The considered biases are the updraft, precipitation, retrievals of AOD and droplet concentration by satellite observations, and vertical co-location of aerosols and clouds. The sensitivity of the cloud droplet number is retrieved and quantified considering different regimes constrained for the potential biases individually.

The paper addresses relevant scientific questions within the scope of ACP and presents ideas using pre-existing data and methods but used for new scientific questions which I find particularly interesting. It presents different conclusions on the potential biases when studying the aerosol impacts on liquid cloud properties. The results are interesting for the scientific community and the paper indicates the credit to related work and the motivation of their new contribution. The abstract reflects the contents of the paper.

However, I have some concerns that need to be addressed. I was hoping for an S value with and without constrains for all the biases to evaluate the impacts all together, it would help to motivate future studies. I think the method section needs more details: There are many datasets used and many constrains, but it is difficult to understand what is used for each section. Considering the lack of my understanding in the method, I cannot assess that the results are sufficient to support the conclusions and it would be difficult for fellow scientists to reproduce the study. Also, the study refers to the Twomey effect, but the authors are not looking at the changes in radiative properties. I find the term of Twomey effect in the title and through the text misleading. Most of the discussions are in the section "results" and not "discussion" and the "discussion" section is more an outlook, but otherwise the paper has a good structure. For all the different reasons mentioned above, I suggest major revisions before accepting the article for publication. I described below the different comments that I think are needed to improve the paper.

We thank the reviewer for the thorough assessment, and helpful comments and suggestions to improve the manuscript. We have revised the manuscript carefully according to the reviewer's comments. Please see the following detailed point-by-point responses.

**Major revisions**

1. As suggested in my introduction, the study claims to deal with the Twomey effect, but the Twomey effect refers to the change in cloud radiative properties due to CCN. The change in cloud droplet number is one of the causes. The cloud droplet concentration can be linked to the Twomey effect only if the water content is maintained constant. I do not think the present study is constraining for water content therefore the change in cloud droplets cannot be related to the Twomey effect here. I understand that biases on S would impact the quantification on the Twomey effect but claiming that "the measure of the Twomey effect (S)" on line 378 is wrong. I suggest limiting the reference to the Twomey effect as a motivation and removing it from the title because it is not what the paper is quantifying.

We thank the reviewer for this important suggestion. For studies that look at the response of cloud effective radius to aerosol, the constraint on LWP is critical. The Nd calculation is, however, independent of LWP (at least in an ideal case), so does not need an explicit constraint to be related to the Twomey effect, which was well documented by McComiskey and Feingold (2012). We agree with the reviewer that the Twomey effect also includes the radiation response to changed aerosol (radiative forcing), but this radiative component is not estimated here as the anthropogenic perturbation to CCN concentrations is highly uncertain and not easily accessible from observational data. Instead, this study places the focus on the microphysical component of the Twomey effect (i.e., $S$), which is central to the overall calculation. We now make this clear on both abstract and introduction (lines 2-3 and lines 31-33). As such, we think it is still useful and meaningful to mention the Twomey effect as a motivation, but meanwhile we limit the cases where $S$ is referred to as they Twomey effect throughout the main text as suggested by the reviewer.

2. I have several problems with the method in section 2:

1) There are a lot of datasets, but I am still confused about which dataset have been used for which part of the study. Maybe it is ok in the section, but I suggest to explicit the datasets used for each part of section 3 (as it is done for section 3.2 on lines 215-218).

   Thank you for the suggestion. To address this problem more clearly, we now add Table 2 to summarize the combination of datasets used in each section.

2) Line 112: I do not understand how aerosol and cloud properties are collocated, since the aerosols are not retrieved if the pixel is cloudy. Can the authors explicit how they deal with that? The closest pixel is mentioned in one of the sections, but I do not know if this is the method for the entire study.

   Since passive remote sensing only allows us to retrieve aerosol properties in clear pixels, in fact we cannot obtain strictly collocated aerosol/cloud retrievals at a cloud pixel scale (e.g., 1 km x 1 km). The only feasible way is to collect adjacent aerosol and cloud retrievals for analysis. Therefore, we use AOD on a coarse-resolved grid ($1° \times 1°$ on a latitude–longitude grid) to match cloud pixels (1km x 1km), assuming that aerosols properties in adjacent clear areas are representative of those under cloudy conditions (Anderson et al., 2003; Quaas et al., 2008). We now make this clear in the method section on lines 137-140.

3) When MISR and Terra are used, I am guessing that A-train observations are not used, but I am not sure. Can the authors clarify?

   Yes, MISR and MODIS/Terra are used in section 3.1, 3.3 and 3.4, and A-train observations are used in section 3.2, which has been clarified in Table 2.

4) Line 117: "the lowest 15%", is this threshold based on Ma et al. (2018) or is it an ad-hoc choice? If the latter, the authors should specify why 15% is chosen. Also, the lowest 15% can be important as it represents a transient mode and potentially the highest impact of aerosols on cloud properties. Did the author study how it potentially impacts the results?

   This threshold is proposed by Hasekamp et al. (2019) based on the finding by (Ma et al., 2018) that the large measurement uncertainties at low aerosol concentrations can lead to an underestimate of cloud susceptibility. (Hasekamp et al., 2019) further showed that leaving out the lowest 15% of data yields a higher $N_d$-to-AI slope compared to using all data, which is also found in our study. A statement on this is added to the revised manuscript (lines 144-146).

5) Line 151: about dividing the dataset into 20 bins of AI/AOD, considering the median, and doing the fit, I understand the idea but I think the dataset loses a lot of information doing that. Also, if the dataset is large enough, the outliers will be removed since most of the points are going to be around the correct value. A statistical test is better than considering the medians. The difference between the blue and white lines in the example (Figure 1) are very similar and does not convince me that the method chosen by the authors is better than considering every data point.

   Thank you for the important comment. Actually, both approaches involve different degrees of averaging (or binning). For instance, even for the all-data approach, each data point (1° by 1° grid) is a result of averaging from sub-grid observations. As the two approaches are the typical ones utilized by previous studies to investigate aerosol-cloud interactions (Quaas et al., 2008; Gryspeerdt et al., 2017; Hasekamp et al., 2019; Rosenfeld et al., 2019), it is of importance to know how large the difference in S estimates between the two could be. Therefore, we rephrase this paragraph to focus more on the difference (lines 201-203) rather than judging which approach is better, which is beyond the scope of this paper. We note that both approaches lead to similar conclusions, as such, we only focus on the results from pre-binned approach in the revised manuscript, meanwhile also put the results associated with all-data approach to Supplementary Materials. The 95 % confidence intervals of the regression slopes are now added for both approaches as suggested by the reviewer.

3. The authors are mentioning the uncertainties from space-based observations and how to reduce them (e.g., looking at certain solar zenithal angles...), but I doubt that the uncertainties are reduced to 0. However, remaining uncertainties are neither included in the results nor discussed by the authors. I think a discussion is missing about the uncertainties in the retrievals but also on the methods: The authors could have retrieved uncertainties on S with the 95% confidence interval on the fit considering the entire dataset (and not only the medians over the 20 bins) as it is done by many previous studies on the aerosol impacts on liquid clouds. Considering uncertainties on the fit would have been relevant when comparing the sensitivity values, for example in lines 222, the difference between 0.45 and 0.56 in S might not be statistically significant, I am not convinced by the comparison made if uncertainties are not provided.

Thank you for the important comment. We now calculate the 95% confidence intervals on the fits throughout the paper (see the response above), and also add a paragraph to discuss the remain uncertainty on $N_d$ retrieval on lines 422-427.

4. I do not understand for which type of clouds the study is designed.

This study was not limited to a specific cloud type. All types including stratiform and cumuliform clouds were considered together in the original manuscript. In order to carefully apply CBH as a updraft proxy, we now constrain clouds to convective clouds but only for the CBH-related results (see more details in response below).

1) The CBH is used as a proxy for the updraft and seems to be designed for cumuliform clouds as stated in lines 55 and 169 but in the method section, I do not see any constrain for avoiding other type of clouds. Therefore, I am wondering if the proxy for the updraft is relevant in most of the observed pixels.

Thanks for this important comment. To ensure the applicability of CBH. we now restrict the analyses related to CBH to convective clouds only by adopting the threshold of LTS < 16K (Rosenfeld et al., 2019). A similar dependence, i.e., increasing S with CBH, is also seen even after constraining the cloud type (Fig. 2a in the revised manuscript). It is worth mentioning that, beyond the CBH, the CGT was used as an alternative proxy for the updraft regardless the cloud types, since it has been observed to be associated with the cloud-base updraft for shallow cumuliform clouds (Lareau et al., 2018) and also correlated with cloud-base updraft for stratiform clouds via modulating cloud top cooling (Zheng et al., 2016). Therefore, the conclusion of updraft-dependency holds for all cloud types. We now clarify these in the method section on lines 160-167, and the Figure 2a is revised accordingly.

2) There are some specifications referring to adiabatic situation (line 120 "adiabatic approximation"), to cumuliform clouds (line 55), or to convective clouds (line 169), but there is no constrain for parameters to limit the study to these situations. Therefore, I am not sure that the proxies used by the study are relevant. There is a threshold on considering single layered clouds (line 123), but I am not sure it is enough.

Thanks for the comment. In the revised version, we have already constrained clouds involved in the CBH-related analysis to convective ones (see response to specific point above). As for the adiabatic approximation, it was found that the filtering of cloud adiabaticity only has a negligible impact on the estimate of S since $N_d$ is the independent variable in the S calculation, but in turn results in a reduction of up to 63 % in the data volume (Gryspeerdt et al., 2021). For this reason, we do not apply such filtering here. We now explain this more carefully in the revised text on lines 149-151.

3) Some results are associated to "stratus clouds" (line 177). I guess that there is no threshold on the type of considered clouds, but then I do not understand how to use linear correlation only applicable to convective clouds and the discussion is about effects from stratus clouds. The authors should clarify that.

The cloud type constraint has been applied (see more details in response above) as suggested by the reviewer. With respect to the discussion involved "stratus clouds", we didn't mean the comparison with previous studies for same cloud type, but wanted to say that the similar dependence on updraft was also observed in different cloud regimes (stratus and also altocumulus clouds) by surface remote sensing.

4) Another example on line 278, where the authors compare DELTANd and DELTANall, these values might be retrieved for different clouds, for open/closed cells, convective clouds, I do not understand how these two quantities can be compared without further consideration on the type of clouds. The differences in S can be explained by the biases, as suggested by the authors, but also by different cloud types, meteorology... The authors acknowledge it on line 410 "CF also covaries with cloud dynamics". Can the authors explain how they can certify that the observed differences are due to the biases and not due to different environments?

Thanks for the comment. For each $1° \times 1°$ cloud scene, we calculated $N_{dAll}$ and $N_d$ concurrently, which ensures same environmental conditions (such as cloud type and meteorology) when comparing the two quanities; thus their difference only reflects the role of retrieval errors. This is now clarified on lines 315-319.

5) There is a cloud regime dataset from MODIS observations, did the authors try to use that to separate the different effects? (Naeyong Cho, Jackson Tan and Lazaros Oreopoulos, L. (2021), MODIS Cloud Regime Level-3 Daily 1 deg x 1 deg, Goddard Earth Sciences Data and Information Services Center (GES DISC), 5067/MEASURES/MODISCR/EQANGD/DATA301)

Thank you for bring this dataset to our attention. Actually, the estimation on sensitivity by cloud regime is also not the focus in this study, and it has already been done by previous study(Gryspeerdt and Stier, 2012). Thus we are not going to repeat it. To select convective clouds, we adopt the threshold of LTS < 16K (Rosenfeld et al., 2019) in the revised version (see response above).

5. I understand that the authors cannot study all the potential biases on the aerosol-cloud interactions, but I am wondering how did they chose? They considered the updraft velocity, but another important meteorological parameter is the humidity for which important effect on S has been demonstrated by previous studies.

Thanks for the comment. A discussion on other important meteorological parameters, including humidity as mentioned by the review, are now added to the introduction (lines 48-52). The reason we choose the updraft is that it determines cloud development as well as the maximum supersaturation at cloud base, and thus how many aerosols can be activated, which is the primary determinant of S (Quaas et al., 2020). Moreover, our understanding on its effect on S is not as sufficient as the humidity due to the difficulty in observing vertical velocity at a global scale. Thus, we placed our focus on the updraft instead of the humidity.

6. Some pixels can be mixed phase clouds but detected as purely liquid by the algorithm, impacting the effective radius, optical thickness, and Nd. I do not know the temperature range on the study, but does the dataset have liquid pixels potentially contaminated by ice?

Thanks for the comment. As noted in the original manuscript (now line 155), in addition to the flag of liquid phase, we also use the criterion of cloud top temperature (CTT) at 1x1 km2 resolution higher than 268 K to avoid the potential contamination of ice pixels (Bennartz and Rausch, 2017). As shown in Fig. R1, the CTT ranges from 268 K to 300 K. Moreover, MODIS-derived CTT was found to underestimate aircraft observations (King et al., 2013); thus it seems unlikely that the selected liquid clouds here are contaminated by ice pixels.

[Figure]

Figure R1: Probability distribution function (PDF) of cloud top temperature in this study.

7. Result section, there are many discussions on the result section which should belong to the discussion section (e.g., from line 176 to line 182, from line 184 to line 191, from line 204 to line 213, from line 237 to line 244, from line 331 to line 339).

Thanks for the comment. The reviewer is right that the 'Discussion' section in the original manuscript is more an outlook, since we wanted to discuss the caveats and suggest potential ways forward here. This section is now renamed as 'Future improvements' to circumvent misleading. The sentences mentioned by the reviewer are more relevant to the specific results and tightly linked to specific figures being discussed in the result section; thus we feel like it would be better to put them in their original place in the text. Actually, concluding discussions on the overall results were already placed in the 'conclusion' section (in the original manuscript) that is now renamed as 'Conclusions and discussions'.

8. The authors decided to study different biases on the aerosol-cloud interactions separately and I am wondering if the biases are not correlated with each other. For example, on section 3.2, the impact of precipitation on S is highlighted but it could also be due to a correlation between precipitation and the CBH (or CGT). Did the authors try to study the effect of precipitation on S constraining for CBH and/or CGT for example? Same apply for the other biases.

The reviewer has a good point. We agree that the sources of bias could be also correlated with each other. However, given the impossibility to combine all datasets used in different sections together (e.g., the CBH/CGT from Terra are observed at 10:30 but the precipitation from Aqua at 13:30 local solar time), we are unable do such constraining with currently used datasets. But we are now planning to make use of CALIOP/CloudSat satellite observations, which provide simultaneous retrievals of aerosol extinction profiles, precipitation, and cloud base height (Mülmenstädt et al., 2018), such that an analysis accounting for all potential sources of bias can be performed. A corresponding statement is added on lines 462-467.

9. Section 3.3 I am confused by this section, and I am not sure to understand the results and the discussion about it, can the authors rephrase this section?

Rephrased as suggested.

1) I am skeptical about looking at aerosols next to clouds in general: The presence of a cloud means that the conditions are different than where there is clear sky. How can the authors make sure that the aerosols, meteorological parameters are the same between clear sky and cloudy sky?

In order to obtain 'co-located' aerosol-cloud retrievals for analysis, the often adopted choice is a $1\circ\times1\circ$ gridding, within which sub-grid clear-sky and cloudy pixels co-exist (if clouds are not fully overcast) and are used for retrieving cloud and aerosol properties, respectively. Within the $1\circ\times1\circ$ grid, aerosol concentrations are considered homogeneous (Anderson et al., 2003) so the clear-sky aerosol concentration is representative of that under the cloudy condition. A corresponding statement is added on lines 305-308. Also note that the same meteorological conditions are not required when talking about the representativeness of aerosol, except for relative humidity, which is associated with aerosol swelling. Actually, the swelling effect has been considered by using the metric $\Delta L$.

2) Also, the authors mentioned that studies on 3D effect and aerosol swelling next to observed clouds are lacking but there are two references about this subject that I think are relevant to this subject and do not appear on the present article: 1. https://doi.org/10.5194/acp-17-13151-2017 2. https://doi.org/10.5194/acp-17-13165-2017

Thanks for these two important references, which are now added. We also rephrase the associated sentences (lines 300-303).

10. Line 264 "It is also noted that SAOD shows first an increase and then decreases from the second DELTA L bin", I do not see what the authors are refereeing to, or is it from the third bin?

We thank the reviewer for the attentive reading of the text. Yes, we meant the third bin. This is now corrected.

11. Line 276 "highly depends on the retrievals bias in clouds", since there is no information on method biases, I believe this statement is too strong.

Thanks for the comment. The method biases have been already inferred by adding the 95% confidence interval as suggested by the reviewer.

12. Line 307, "The strength of the Twomey effect derived on a basis of column-integrated aerosol quantity. . .", I am confused because on line 294, the authors said that SO4C AND SO4S mimic the column integrated, and here only SO4C shows a large slope so how is it directly linked to this? and SO4S does not show the trend described on line 307.

In this study, we use SO4C to mimic the column integrated aerosols (e.g., AI/AOD) but SO4S to mimic the surface aerosol quantities (e.g., aerosol extinction coefficient). This sentence has been rephrased to avoid misleading.

13. Along the article, different methods are employed to explain the different biases, I was expecting at the end a value of S considering all the possible biases, (precipitation, too close to the cloud, ...) for latitude bands for example and/or season, but no. Each paragraph is developed individually and at the end the discussion does not bring all of them together.

This is a very good suggestion by the reviewer. See response to specific point above(MajorRevision#8).

14. Figure 1:

1) Why the blue and white lines do not go until AI=1. The authors mention that the lowest values of AI are removed, but they do not refer to large values of AI (AI>0.5).

Thanks for the reminder. We did not remove the large values of AI. This is just a plotting thing, and has been corrected.

2) Also, in the plot of PDF Vs AI, I do not understand why the PDF is almost equal to 0 for AI 0.5 whereas on the upper plot the PDF of the data are clearly greater than the 0 (might even be higher than the maximum at AI 0.8).

As noted in the original manuscript (the caption of Fig. 1), the upper plot is a joint $N_{\mathrm{d}}$–AI histogram, where each column is normalized so that it sums to 1. That is, this plot shows the probability of finding a specific $N_{\mathrm{d}}$, given that a certain AI has been observed, instead of the occurrence frequency distribution of all data.

3) I think it would be better to indicate the value of the regression to see the difference between the blue and white lines and discuss about it, there is nothing about it in the data and method section.

Thanks for the suggestion. The regression slopes (with 95% confidence interval) are now added to the plot and discussed (lines 203-205), and data used to generate this schematic diagram are also clarified in the caption.

4) Why the study uses the blue line instead of the white lines? The authors could infer the uncertainty through the 95% confidence interval using the white lines.

Added. See response to specific point above (MajorRevision#2.5).

**Minor revisions**

1. All along the text, there are several words which are unnecessary in my opinion (e.g., line 112 "basically", line 175 "It is clear that", line 175 "evident", line 183 "remarkably", line 219 "as expected", line 220 "much", line 226 "evidently", line 229 "obviously", line 232 "clearly", line 271 "serious", line 275 "sharply", line 293 "practically", line 319 "much", line 341 "clearly"), and sometime they imply that something is evident but it is not the case (in my opinion).

We thank the reviewer for the attentive reading. Modified as suggested.

2. Line 7 "consistent with stronger aerosol-cloud interactions at larger updraft velocity", this is not the case for every type of clouds (e.g., arctic stratus).

Thanks for the reminder. Corrected.

3. Line 25 "This study will", I suggest moving this sentence to the last sentence of the paragraph.

Thanks for the suggestion. Revised.

4. Line 30 "As reviewed recently...", I do not understand in which context the present study is related to Quaas et al. (2020), will they consider the same biases, new ones... I think, the authors could clarify this part in the introduction. It makes sense afterward but not reading the introduction for the first time.

Thanks for the suggestion. Quaas et al. (2020) is a recent review paper that summarized current challenges and issues obscuring an accurate estimate of the Twomey effect from satellite observations. Building on this review, we investigate several understudied aspects, which are important but not yet understood in a clear way as reviewed by (Quaas et al., 2020). We have added a sentence to make this clear (lines 57-58).

5. Line 55 "their strong correlation illustrated by in-situ observations of cumuliform clouds", this sentence is important as it is a key correlation used through the study, so I think the authors could elaborate a bit more on the limitations of it.

Thanks for the comment. We write a statement on this now in the revised manuscript (on lines 68-71).

6. Line 116 "a standard deviation higher than the mean value", does this threshold come from somewhere specific?

The threshold stems from the study by Saponaro et al. (2017), which has been cited in the main text.

7. Line 117 "the lowest 15%", is this threshold based on Ma et al. (2018) or is it an ad-hoc choice. If the latter, the authors could specify why 15% is chosen.

See response to specific point above and line 144.

8. Lin 123 (Feingold et al., 2021), it should not have parenthesis.

Thank you. Corrected.

9. Line 140, is there a reason why "MYD06 5-km" is not written "MYD05 5x5 km2" to be consistent with "CloudSat data at a 1.4x2.5km2".

   Thanks for the reminder. Corrected.

10. Line 146, the authors mention that they considered sulfate from MERRA-2, I am wondering if they considering other species.

    Thanks for the comment. Other species are not considered in this study, because variability in sulfate aerosols has been found to contributes the most strongly to variability in $N_d$ among all aerosol species (McCoy et al., 2017). We now clarify this on lines 185-187.

11. Line 162 "Nd is essentially a function of both CCN and updraft", a citation should be added here.

    Thank you. Added on line 210.

12. Line 190 "Sai is consistently higher than SAOD", not always as for CGT 900m

    We thank the reviewer for this attentive reading of the text. Indeed, it is not the case for CGT 900m. We now revise the text accordingly.

13. Line 195 "They proposed...", Who are they? Are they Reutter and al.? If so, they should not be put in parenthesis in the sentence before.

    Yes, 'they' are Reutter et al. Thanks for the comment. Corrected on line 249.

14. Line 199 "proxy of updraft (CBH/CGT)", are the quartiles enough to discriminate the updraft regimes described by Reutter et al.? The quartiles defined regimes based on how likely an updraft regime occurs defining different regimes, but I am not sure that they separate in the regimes described by Reutter et al., I am not sure we are in category b.

    To check the robustness of the assumption, we further constrain the variation of CGT to a smaller range (i.e., < 10th and > 90th percentiles), and very similar results are obtained (Fig. S2). With the quasi-constant CGT, AI can thus be assumed as an indicator of regime. We now add a statement on this (lines 248).

15. Lines 201-203, "As illustrated in the ..." I am confused by this sentence; can the authors rephrase this sentence? What I understand:

    1) At low AI, the updraft should have limited impact on Nd (case a from Reutter et al.), but looking at the plot 3c and i, the updraft has a strong impact.
       Yes, at low AI, what should be expected is the similar distribution of $N_d$ between different cloud dynamics as determined by the nature of aerosol-limited regime, or at least a slightly higher $N_d$ for the strong updraft case. However, looking at the clean zone (AI < 0.15) in Fig. 3, the strong updraft is associated with much lower $N_d$ as well as larger CER (generally larger than 14 μm, the threshold for drizzle initiation) compared to the weak updraft. Note that this is not relevant to the role of strong updraft facilitating activation of cloud droplets but a possible role of precipitation and/or drizzle. This has been clarified in the original manuscript (now lines 258-263)

    2) On the opposite, at high AI, the updraft should most likely have a strong impact, but looking at the plot, the values are messier, and I do not observe a strong dependence on Nd with AI here. Can the dependence be quantified by the authors?
       It is known that the impact of aerosol on cloud is strong at low aerosol concentration, whereas a saturation effect occurs as the aerosol keeps rising. Thus the dependence of Nd on AI is supposed to be weak at high AI as we see.

    3) Maybe it is irrelevant, but I am wondering why the authors based their regimes on CGT and not the ratio CGT/AI?
       Thanks for the question. Since our focus is to contrast AI-$N_d$ relationships at strong (high) and weak (low) updrafts (CGT), it is more straightforward to illustrate AI-$N_d$ joint histogram for constrained CGT intervals. By doing so, the information of regimes can be also inferred via AI simply.

16. Line 228 "appears to strengthen the aerosol-Nd relationship", can the author develop a bit more on this?

    This is a speculation made by (Painemal et al., 2020) to explain their observed higher aerosol-$N_d$ correlation for all clouds compared to non-drizzling clouds. Unfortunately, they did not dig more deeply.

17. Section 3 "AOD", the authors mentioned earlier in the text that they would not consider AOD and prefer the use of AI, but they use AOD in this section. Why changing the considered parameter?

    Thank you for this question. We know that AI is not directly retrieved but calculated from AOD, so it is necessary to look at AOD as well when discussing retrieval problems. We now make this clear on lines 321-322.

18. Lines 264 to 268, I do not understand this part, can the authors rephrase it?

    Rephrased as suggested (lines 331-337).

19. Line 282 "it is clearly illustrated that CF regulates the negative correlation between DELTAL and DELTANd", can the authors provide more information on that part?

    The detail on how CF regulates the negative correlation have been added on lines 354-356.

20. Line 287 "Given that CF also correlates closely with cloud dynamics", the correlation presented by the authors are based on the medians which are not very significant in my opinion, 2D histogram and regression on the entire dataset would be better.

    Thanks for the comment. The joint histogram between CF and CGT is now added to Fig. 6 as suggested. Given that the relationship is highly non-linear, a singe regression slope would not make much sense; such slope is thus not included.

21. Line 295 "commonly used", can the author support this with citations? Maybe some references on the use of satellite observation combined with models to study aerosol cloud interactions are missing here.

    "Commonly used" here refers to AOD/AI and surface aerosol extinction coefficient from satellite-based and ground-based observations rather than SO4C and SO4S from reanalysis data. Although not as commonly adopted as AOD/AI and surface aerosol extinction coefficient, SO4C and SO4S were also used as CCN proxies by previous studies (McCoy et al., 2017; Jia et al., 2021). This sentence has been revised to avoid misleading, and the relevant references have been cited as well.

22. Line 296 "is considered to be more relevant to the amount of CCNs", can the author provide a citation for this statement?

    Thank you. Added on line 371.

23. Line 302 "pre-binned method", is it the method described in Figure 1? If this is the case, can it be explicit? I am still skeptical, and I would prefer statistics on the entire dataset and not on the medians (as done in Table S1).

    Clarified. As replied above, the analyses on both pre-binned data and entire data are included in the revised manuscript.

24. Line 302 "binend"-> "binned".

    Corrected.

25. Line 320 "such as western North Pacific and the Atlantic", this is true for SO4B but not for SO4C (also East coast of south America and South Africa), or maybe I am misunderstanding.

    The reviewer is right that high CVs are also evinced at East coast of south America and South Africa. We revise the text accordingly (line 397).

26. Line 321 "the spatial CV of SO4C exhibits a much smaller (0.88) value than those of SO4B and SO4S (1.84 and 1.79)", are these values averaged over North Pacific and Atlantic, if this is the case the authors should specify the limit in lat/lon of the considered box.

    The values of spatial CVs are calculated from the global geographical distribution instead regional one. This is now clarified on line 394.

27. Line 330 "loose correlation (R<0.3) . . .", can the authors quantify or rephrase that because I am not convinced especially for r(SO4B, SO4S) which seems high when the ratio is small.

    Thanks for the suggestion. We now rephrase this sentence as suggested on line 408. Also note that this sentence is only relevant to r(SO4B,SO4C) instead of r(SO4B,SO4S).

28. From line 331 to line 339, I find the conclusions on this discussion very strong. I think it should be at least quantified to support that.

    Modified as suggested.

29. Line 406 "In terms of aerosol. . .", is this sentence part of point 3 or point 4?

    This is a part of point #3. Here, the use of aerosol reanalysis is recommended to avoid retrieval biases in AOD/AI, while in the point 4, the use of reanalysis is to address the problem of vertical co-location.

30. Line 419 "SO4B-S-C", can the authors specify again on the different quantity in the conclusions?

    Thanks for the reminder. The meanings of SO4C, SO4B, and SO4S have been specified in the conclusion section on lines 506, 507, and 513.

**References**

[revised manuscript text omitted]

---

## Author Comment (AC4)

**Response to the Reviewer's Comments**

March 17, 2022

We would like to thank the reviewers for the effort in helping us improve the manuscript. Below we respond point-by-point to the comments, with the reviewer comments in black, and our responses in blue. The manuscript has been revised accordingly. The line numbers in the response are for the revised version with tracked changes.

**Comments by Reviewer #1**

This study investigates the changes of Twomey effect for marine warm clouds with various influential factors including the updraft, precipitation, retrieval errors, and vertial co-location between aerosol and clouds. Valuable results have been obtained, which can improve our understanding of the radiative impacts from aerosol-cloud interaction from perspective of satellite observations. Also, the paper is well written. I personally think this manuscript is suitable for publication after a minor revision.

We thank the reviewer for this thoughtful and constructive statement. We have revised the manuscript carefully according to the reviewer's comments. Please see the following detailed point-by-point responses.

**Detail comments**

1. Line 19-21, In addition to the radiative impacts of aerosols by serving as CCN, aerosols can also affect the development of clouds and then precipitation and radiation by modifying the thermal structure of atmosphere via direct radiative effect.

   Thanks for the comment. This effect is now included in the introduction on lines 23-24.

   *Added text:*

   *Lines 23-24: 'Additionally, absorbing aerosols can also alter the cloud distribution by perturbing the atmospheric temperature structure, known as semi-direct effects (Allen et al., 2019).'*

2. Line 24-25, regarding the rapid adjustments, one reference is suggested here which showed the increase of cloud liquid water path and decrease of cloud re with incrased Nd via Twomey effect, Zhao and Garrett (2015, doi: 10.1002/2014GL062015).

   Thanks for bringing this study to our attention. We now cite this paper in the revised text (line 29).

3. Line 28-29, Actually even with the same climate model simulation (such as CAM5), the aerosol first indirect effect also varies with the aerosol variables that are used to present the aerosol amount.

   We agree with the reviewer that the choice of aerosol proxy can play a critical role. This point was discussed in the original manuscript (now lines 41-42).

4. Line 33-41, There are various influential factors, which are not limited to these five points. For example, the existence of precipitation particles within clouds as indicated by Yang et al. (2021, doi: 10.1029/2021JD035609) based on satellite observations, the aerosol amount or availability of water vapor amount as indicated by Qiu et al. (2017, doi: 10.1016/j.atmosenv.2017.06.002), cloud types or vertical locations as indicated by Zhao et al. (2019, doi: 10.3390/atmos10010019), and potential large uncertainties in cloud retrievals as indicated by Zhao et al. (2012, doi: 10.1029/2011JD016792), and so on.

   Thanks for the comment. These influential factors have been discussed in the introduction and the relevant references have been cited.

*Added/Modified text:*

*Lines 49-53: 'Additionally, meteorological conditions, e.g., lower tropospheric stability (Ma et al., 2018), relative humidity (Quaas et al., 2010), availability of water vapor (Qiu et al., 2017), and wind shear (Fan et al., 2009), and vertical overlapping status of aerosol and cloud layers (Costantino and Bréon, 2013; Zhao et al., 2019) also play roles in regulating aerosol-cloud interactions. It is worth noting that most of these studies calculated S based on cloud effective radius rather than $N_d$, and so are subject to even more errors from the problem of stratification by liquid water path.'*

*Lines 180-181: 'As a sink of $N_d$, drizzle could also affect the aerosol-cloud interactions even without rain falling on ground (Yang et al., 2021), so we also include drizzling clouds into precipitating cases. '*

*Lines 88-91: 'In terms of $N_d$, retrievals for 3-D-shaped clouds and partially cloudy pixels deviate from the retrieval assumptions of overcast homogenous cloud and 1-D plane-parallel radiative transfer, thereby appear to lead to an overestimation of CER (Coakley et al., 2005; Matheson et al., 2006; Zhang and Platnick, 2011; Zhao et al., 2012), in turn, an underestimated $N_d$ (Grosvenor et al., 2018).'*

5. Line 54-56, Good idea. However ,with this assumption or method, we may limit the cloud types as cumuliform clouds.

   We thank the reviewer for this very important comment. To ensure the applicability of CBH as a updraft proxy. we now restrict the analyses related to CBH to convective clouds only by adopting the threshold of LTS<16 K (Rosenfeld et al., 2019). A similar dependence, i.e., increasing S with CBH, is also seen even after constraining the cloud type (Figure 2a in the revised manuscript). It is worth mentioning that, beyond the CBH, the CGT was used as an alternative proxy for the updraft regardless the cloud types, since it has been observed to be associated with the cloud-base updraft for shallow cumuliform clouds(Lareau et al., 2018) and also correlated with cloud-base updraft for stratiform clouds via modulating cloud top cooling (Zheng et al., 2016). Therefore, the conclusion of updraft-dependency holds for all cloud types. We now clarify these in the method section on lines 160-167, and the Figure 2a is revised accordingly.

   *Added/Modified text:*

   *Lines 160-167: 'To overcome the lack of the global updraft observation, we utilize satellite-based retrievals for CBH as a proxy of cloud base updraft for cumuliform clouds, based on the finding that these two quantities exhibit an approximately linear correlation for convective clouds (Zheng and Rosenfeld, 2015). Here, clouds are considered convective for low troposphere static stability (LTS) less than 16 K (Rosenfeld et al., 2019). Additionally, cloud geometrical thickness (CGT; the difference between cloud top height and CBH) is used as an alternative proxy for the updraft regardless cloud types, since it has been observed to be associated with the cloud-base updraft for shallow cumuliform clouds (Lareau et al., 2018) and also correlated with cloud-base updraft for stratiform clouds via modulating cloud top cooling (Zheng et al., 2016). '*

6. Line 65-66, The reference mentioned above (Yang et al. 2021) also took use of the simple threshold value method with 14 um.

   Thank you. The paper is now cited in the revised manuscript.

7. Line 69, Do the authors mean "Solving this problem is helpful to ..."?

   Thanks for the comment. We have rephrased the sentence as suggested by the reviewer.

8. Line 77-81, Actually, the existence of aerosols could also cause biases to satellite-based cloud retrievals. As indicated by Li et al. (2014, doi: 10.1002/2013JD021053), the existence of absorbing aerosols could cause the satellite based retrieval of optical depth lower, effective radius higher, and so on.

   Thanks for the comment. We now cite this paper and discuss the retrieval uncertainty on lines 94-95.

   *Added text:*

   *Lines 94-95: 'In addition to the assumptions on clouds, the existence of aerosols above clouds can also affect the retrieval of cloud optical depth (Haywood et al., 2004; Li et al., 2014), in turn bias $N_d$ calculation.'*

9. Line 107-110, previous studies have already indicated that the aerosol-cloud interaction is sensitive to the spatial resolution. How do the authors consider this point?

Thanks for this comment. The reviewer is right that the varying spatial resolution can also play a role. We did not explicitly consider this issue since this is not the focus of this study, but a discussion on this issue is now added on lines 130-133.

*Added text:*

*Lines 130-133: 'It is worth mentioning that, as S was found to vary with the spatial resolution of data (Sekiguchi et al., 2003; McComiskey and Feingold, 2012), the different data resolutions between section 3.2 and other sections can lead to a difference in S; but this is not the focus here.'*

10. Line 110-111, It is well known that the retrieval uncertainties are large over polar regions, how about that over land regions? A reference might be helpful.

    Thanks for the suggestion. We do this in the revised manuscript (line 134).

    *Modified text:*

    *Line 134: 'This study is restricted to global ocean with latitude between $60°S$ and $60°N$ because of limited quality of retrievals of aerosol size parameters (Levy et al., 2013) and $N_d$ (Gryspeerdt et al., 2021) over land and polar regions.'*

11. Line 112-125, why are the Level 3 aerosol data but Level 2 cloud data used in this study?

    This is because the use of L2 cloud data can avoid the aggregation bias of $N_d$ calculation (Feingold et al., 2021), and also enable us to select more reliable cloud retrievals on the level of the satellite pixel (1x1 km2) with different flags as detained on lines 154-159. These reliable cloud retrievals are then aggregated to larger scales to match Level 3 aerosol retrievals.

12. Line 123, it might be better used as Feingold et al. (2021)

    Thanks for the reminder. Corrected.

13. Line 130-136, what are the potential limiations or uncertianties in the cloud base height retrievals by the introduced method? It is worthy to briefly describe.

    The best performance of this algorithm is achieved for clouds with CBH around 1 km and CGT below 1 km. For such heights, which are characteristic for oceanic clouds considered in this analysis, the root mean square error ranges between 300–350 m. It is important to note that the MISR cloud-base height retrieval is limited to CBH > 560 m (Böhm et al., 2019). At this lower end of the detection range, a slight underestimation of the CBH is expected (Böhm et al., 2019). This is now clarified on lines 170-174.

    *Added/Modified text:*

    *Lines 170-174: 'The best performance of this algorithm is achieved for clouds with CBH around 1 km and CGT below 1 km. For such heights, which are characteristic for oceanic clouds considered in this analysis, the root mean square error ranges between 300–350 m. It is important to note that the MISR cloud-base height retrieval is limited to CBH > 560 m (Böhm et al., 2019). At this lower end of the detection range, a slight underestimation of the CBH is expected (Böhm et al., 2019). '*

14. Line 136-141, even within non-precipitating clouds, drizzle could exist and affect the aerosol-cloud interaction, as indicated by the reference mentioned earlier Yang et al. (2021), how could the authors consider this impact?

    Thanks for the question. As the drizzle affects the aerosol-cloud interactions in a same manner with rainfall, i.e., as a strong sink of $N_d$, we considered both 'liquid precipitation' and 'possible drizzle' as precipitating clouds (see line 179). The important result of Yang et al. (2021) is now mentioned on lines 180-181.

    *Added text:*

    *Lines 180-181: 'As a sink of $N_d$, drizzle could also affect the aerosol-cloud interactions even without rain falling on ground (Yang et al., 2021), so we also include drizzling clouds into precipitating cases. '*

15. Line 180-181, One possibility is the large volume of datasets. Could the data selection also play a role to the higher correlation?

    The reviewer has a great point. It might somehow play a role. However, given the similar cloud type, i.e., ice-free and single layered clouds, selected in both studies, we can not conclude the data selection is the reason for the higher correlation.

16. Line 200-201, it is easy to understand that the low AI zone is more likely aerosol-limited. However, I cannot understand why the high AI zone is close to updraft-limited regime if we do not know how large the updraft is? Could the authors expalin more?

    Thanks for the question. As we have limited the proxy of updraft to a certain range (< the 10th/25th percentile or > the 75th/90th percentile) in Fig.3 and Fig. S2, it is thus assumed that the ratio of updraft to AI is dominated by AI. A statement on this is now added on line 248.

    *Added text:*

    *Line 248: 'Note that applying the 10th and 90th percentiles also yields similar results as shown in Fig. S2.'*

17. Line 226, I would suggest using the same format, either with or without parathesis for ln AI.

    Thanks for the reminder. We now remove the parentheses of ln (AI), ln ($N_d$), ln(SO4C/B/S) throughout the text.

18. Line 233-237, if possible, I personally would like to suggest seperating this long sentence to a few short sentences.

    Thank you for the suggestion. This sentence is now split into three short ones (lines 288-292).

    *Modified test:*

    *Lines 288-292: 'As shown in Fig 4g, the polluted clouds consist predominately of the non-raining clouds as a result of the suppression of precipitation by aerosols, thus maintaining a high value of $N_d$. Instead, the majority of the clean clouds are raining ones that are significantly subjected to the sink processes for $N_d$ and/or aerosol scavenging (Boucher and Quaas, 2013) (Fig. 4f), hence corresponding to a lower $N_d$.'*

19. Line 270, 'appears to'-> 'appear to'

    Thank you. Corrected.

20. Line 317, why do the authors choose to use daily time series values instead of hourly?

    Because aerosol and cloud properties are linked on a daily basis throughout the manuscript, we thus choose daily time series to derive the temporal CV so as to quantify the day-to-day variability of SO4.

**References**

[revised manuscript text omitted]

---

## Author Comment (AC5)

**Response to the Reviewer's Comments**

March 17, 2022

We would like to thank the reviewer for the effort in helping us improve the manuscript. Below we respond point-by-point to the comments, with the reviewer comments in black, and our responses in blue. The manuscript has been revised accordingly. The line numbers in the response are for the revised version with tracked changes.

**Comments by Reviewer #2**

The article studies the effect of aerosols on liquid cloud droplet concentration using several space-based instruments (active and passive) to describe cloud and aerosol properties and reanalysis to retrieve information on sulfate. The authors investigate the potential biases that are usually overlooked by most of the studies quantifying this effect. The considered biases are the updraft, precipitation, retrievals of AOD and droplet concentration by satellite observations, and vertical co-location of aerosols and clouds. The sensitivity of the cloud droplet number is retrieved and quantified considering different regimes constrained for the potential biases individually.

The paper addresses relevant scientific questions within the scope of ACP and presents ideas using pre-existing data and methods but used for new scientific questions which I find particularly interesting. It presents different conclusions on the potential biases when studying the aerosol impacts on liquid cloud properties. The results are interesting for the scientific community and the paper indicates the credit to related work and the motivation of their new contribution. The abstract reflects the contents of the paper.

However, I have some concerns that need to be addressed. I was hoping for an S value with and without constrains for all the biases to evaluate the impacts all together, it would help to motivate future studies. I think the method section needs more details: There are many datasets used and many constrains, but it is difficult to understand what is used for each section. Considering the lack of my understanding in the method, I cannot assess that the results are sufficient to support the conclusions and it would be difficult for fellow scientists to reproduce the study. Also, the study refers to the Twomey effect, but the authors are not looking at the changes in radiative properties. I find the term of Twomey effect in the title and through the text misleading. Most of the discussions are in the section "results" and not "discussion" and the "discussion" section is more an outlook, but otherwise the paper has a good structure. For all the different reasons mentioned above, I suggest major revisions before accepting the article for publication. I described below the different comments that I think are needed to improve the paper.

We thank the reviewer for the thorough assessment, and helpful comments and suggestions to improve the manuscript. We have revised the manuscript carefully according to the reviewer's comments. Please see the following detailed point-by-point responses.

**Major revisions**

1. As suggested in my introduction, the study claims to deal with the Twomey effect, but the Twomey effect refers to the change in cloud radiative properties due to CCN. The change in cloud droplet number is one of the causes. The cloud droplet concentration can be linked to the Twomey effect only if the water content is maintained constant. I do not think the present study is constraining for water content therefore the change in cloud droplets cannot be related to the Twomey effect here. I understand that biases on S would impact the quantification on the Twomey effect but claiming that "the measure of the Twomey effect (S)" on line 378 is wrong. I suggest limiting the reference to the Twomey effect as a motivation and removing it from the title because it is not what the paper is quantifying.

We thank the reviewer for this important suggestion. For studies that look at the response of cloud effective radius to aerosol, the constraint on LWP is critical. The Nd calculation is, however, independent of LWP (at least in an ideal case), so does not need an explicit constraint to be related to the Twomey effect, which was well documented by McComiskey and Feingold (2012). We agree with the reviewer that the Twomey effect also includes the radiation response to changed aerosol (radiative forcing), but this radiative component is not estimated here as the anthropogenic perturbation to CCN concentrations is highly uncertain and not easily accessible from observational data. Instead, this study places the focus on the microphysical component of the Twomey effect (i.e., S), which is central to the overall calculation. We now make this clear on both abstract and introduction (lines 2-3 and lines 31-33). As such, we think it is still useful and meaningful to mention the Twomey effect as a motivation, but meanwhile we limit the cases where S is referred to as they Twomey effect throughout the main text as suggested by the reviewer.

**Added text:**

Lines 2-3: 'The microphysical component of the Twomey effect – cloud droplet number concentration  $(N_d)$ -toaerosol sensitivity (S) – is central to the Twomey effect calculation.'

Lines 31-33: 'This study will discuss the Twomey effect with a focus on the sensitivity of  $N_d$  to CCN perturbations, due to its fundamental role in aerosol-cloud interactions. Note that the related radiative forcing will be not estimated here, as the anthropogenic perturbation to CCN concentrations is highly uncertain and not easily accessible from observational data.'

- 2. I have several problems with the method in section 2:
  - There are a lot of datasets, but I am still confused about which dataset have been used for which part of the study. Maybe it is ok in the section, but I suggest to explicit the datasets used for each part of section 3 (as it is done for section 3.2 on lines 215-218). Thank you for the suggestion. To address this problem more clearly, we now add Table 2 to summarize the combination of datasets used in each section.
  - 2) Line 112: I do not understand how aerosol and cloud properties are collocated, since the aerosols are not retrieved if the pixel is cloudy. Can the authors explicit how they deal with that? The closest pixel is mentioned in one of the sections, but I do not know if this is the method for the entire study.

Since passive remote sensing only allows us to retrieve aerosol properties in clear pixels, in fact we cannot obtain strictly collocated aerosol/cloud retrievals at a cloud pixel scale (e.g., 1 km x 1 km). The only feasible way is to collect adjacent aerosol and cloud retrievals for analysis. Therefore, we use AOD on a coarse-resolved grid  $(1^{\circ} \times 1^{\circ}$  on a latitude–longitude grid) to match cloud pixels (1km x 1km), assuming that aerosols properties in adjacent clear areas are representative of those under cloudy conditions (Anderson et al., 2003; Quaas et al., 2008). We now make this clear in the method section on lines 137-140.

Added text:

Lines 137-140: 'In order to collect co-located (adjacent) aerosol and cloud retrievals for analysis, aerosol retrievals on a coarse-resolved grid ( $1^{\circ} \times 1^{\circ}$  on a latitude-longitude grid) are used to match cloud pixels ( $1 \times 1 \text{ km}^2$ ), assuming that aerosols properties in adjacent clear areas are representative of those under cloudy conditions (Anderson et al., 2003; Quaas et al., 2008).'

3) When MISR and Terra are used, I am guessing that A-train observations are not used, but I am not sure. Can the authors clarify?

Yes, MISR and MODIS/Terra are used in section 3.1, 3.3 and 3.4, and A-train observations are used in section 3.2, which has been clarified in Table 2.

4) Line 117: "the lowest 15%", is this threshold based on Ma et al. (2018) or is it an ad-hoc choice? If the latter, the authors should specify why 15% is chosen. Also, the lowest 15% can be important as it represents a transient mode and potentially the highest impact of aerosols on cloud properties. Did the author study how it potentially impacts the results?

This threshold is proposed by Hasekamp et al. (2019) based on the finding by (Ma et al., 2018) that the large measurement uncertainties at low aerosol concentrations can lead to an underestimate of cloud susceptibility. (Hasekamp et al., 2019) further showed that leaving out the lowest 15% of data yields a higher  $N_{\rm d}$ -to-AI slope compared to using all data, which is also found in our study. A statement on this is added to the revised manuscript (lines 144-146).

Added/Modified text:

Lines 144-146: 'As suggested by Hasekamp et al. (2019), the lowest 15 % of data for AOD (AI) at a global scale are excluded to avoid large retrieval uncertainty at low aerosol concentrations (Ma et al., 2018). Note that leaving out the low AOD (AI) yields a larger S compared to using all data (Hasekamp et al., 2019).'

5) Line 151: about dividing the dataset into 20 bins of AI/AOD, considering the median, and doing the fit, I understand the idea but I think the dataset loses a lot of information doing that. Also, if the dataset is large enough, the outliers will be removed since most of the points are going to be around the correct value. A statistical test is better than considering the medians. The difference between the blue and white lines in the example (Figure 1) are very similar and does not convince me that the method chosen by the authors is better than considering every data point.

Thank you for the important comment. Actually, both approaches involve different degrees of averaging (or binning). For instance, even for the all-data approach, each data point (1° by 1° grid) is a result of averaging from sub-grid observations. As the two approaches are the typical ones utilized by previous studies to investigate aerosol-cloud interactions (Quaas et al., 2008; Gryspeerdt et al., 2017; Hasekamp et al., 2019; Rosenfeld et al., 2019), it is of importance to know how large the difference in S estimates between the two could be. Therefore, we rephrase this paragraph to focus more on the difference (lines 198-203) rather than judging which approach is better, which is beyond the scope of this paper. We note that both approaches lead to similar conclusions, as such, we only focus on the results from pre-binned approach in the revised manuscript, meanwhile also put the results associated with all-data approach to Supplementary Materials. The 95 % confidence intervals of the regression slopes are now added for both approaches as suggested by the reviewer.

**Added/Modified text:**

Lines 198-203: 'Additionally, the linear regression on all data points is also shown (yellow dashed line) in Fig. 1 for comparison with the pre-binned approach, since both approaches have been used extensively by previous studies (Quaas et al., 2008; Gryspeerdt et al., 2017; Hasekamp et al., 2019; Rosenfeld et al., 2019) but it is unclear yet how large the difference in estimates between two approaches could be.'

3. The authors are mentioning the uncertainties from space-based observations and how to reduce them (e.g., looking at certain solar zenithal angles...), but I doubt that the uncertainties are reduced to 0. However, remaining uncertainties are neither included in the results nor discussed by the authors. I think a discussion is missing about the uncertainties in the retrievals but also on the methods: The authors could have retrieved uncertainties on S with the 95% confidence interval on the fit considering the entire dataset (and not only the medians over the 20 bins) as it is done by many previous studies on the aerosol impacts on liquid clouds. Considering uncertainties on the fit would have been relevant when comparing the sensitivity values, for example in lines 222, the difference between 0.45 and 0.56 in S might not be statistically significant, I am not convinced by the comparison made if uncertainties are not provided.

Thank you for the important comment. We now calculate the 95% confidence intervals on the fits throughout the paper (see the response above), and also add a paragraph to discuss the remain uncertainty on  $N_{\rm d}$  retrieval on lines 422-427.

**Added text:**

Lines 422-427: 'The derivation of  $N_d$  from satellite observations relies on a number of assumptions (Grosvenor et al., 2018), making it prone to systematic biases. While some sampling strategies have been applied to sidestep the biases in  $N_d$  retrieval (see section 2), the uncertainties remain. To further ensure the cloud adiabaticity, there are two practical methods for use, including comparing the CER at different wavelengths (Bennartz and Rausch, 2017) and locating cloud "core" (Zhu et al., 2018). Appropriate  $N_d$  sampling strategies are anyway beneficial in future investigations, though it has relatively little impact on S (and the implied  $RF_{aci}$ )(Gryspeerdt et al., 2021). '

4. I do not understand for which type of clouds the study is designed.

This study was not limited to a specific cloud type. All types including stratiform and cumuliform clouds were considered together in the original manuscript. In order to carefully apply CBH as a updraft proxy, we now constrain clouds to convective clouds but only for the CBH-related results (see more details in response below).

1) The CBH is used as a proxy for the updraft and seems to be designed for cumuliform clouds as stated in

lines 55 and 169 but in the method section, I do not see any constrain for avoiding other type of clouds. Therefore, I am wondering if the proxy for the updraft is relevant in most of the observed pixels.

Thanks for this important comment. To ensure the applicability of CBH. we now restrict the analyses related to CBH to convective clouds only by adopting the threshold of LTS

Figure R1: Probability distribution function (PDF) of cloud top temperature in this study.

7. Result section, there are many discussions on the result section which should belong to the discussion section (e.g., from line 176 to line 182, from line 184 to line 191, from line 204 to line 213, from line 237 to line 244, from line 331 to line 339).

Thanks for the comment. The reviewer is right that the 'Discussion' section in the original manuscript is more an outlook, since we wanted to discuss the caveats and suggest potential ways forward here. This section is now renamed as 'Future improvements' to circumvent misleading. The sentences mentioned by the reviewer are more relevant to the specific results and tightly linked to specific figures being discussed in the result section; thus we feel like it would be better to put them in their original place in the text. Actually, concluding discussions on the overall results were already placed in the 'conclusion' section (in the original manuscript) that is now renamed as 'Conclusions and discussions'.

8. The authors decided to study different biases on the aerosol-cloud interactions separately and I am wondering if the biases are not correlated with each other. For example, on section 3.2, the impact of precipitation on S is highlighted but it could also be due to a correlation between precipitation and the CBH (or CGT). Did the authors try to study the effect of precipitation on S constraining for CBH and/or CGT for example? Same apply for the other biases.

The reviewer has a good point. We agree that the sources of bias could be also correlated with each other. However, given the impossibility to combine all datasets used in different sections together (e.g., the CBH/CGT from Terra are observed at 10:30 but the precipitation from Aqua at 13:30 local solar time), we are unable do such constraining with currently used datasets. But we are now planning to make use of CALIOP/CloudSat satellite observations, which provide simultaneous retrievals of aerosol extinction profiles, precipitation, and cloud base height (Mülmenstädt et al., 2018), such that an analysis accounting for all potential sources of bias can be performed. A corresponding statement is added on lines 462-467.

**Added text:**

Lines 462-467: 'Given the impossibility to combine all datasets used in different sections together (e.g., the CBH/CGT from Terra are observed at 10:30 but the precipitation from Aqua at 13:30 local solar time), this work evaluates the individual impact of each bias on the estimate of S separately. Nevertheless, the sources of bias could be also correlated with each other; thus an optimal estimate of S with all biases constrained is desirable. Future studies are being planned to make use of CALIOP/CloudSat satellite observations, which provide simultaneous retrievals of aerosol extinction profiles, precipitation, and cloud base height (Mülmenstädt et al., 2018), such that an analysis accounting for all potential sources of bias can be performed. '

9. Section 3.3 I am confused by this section, and I am not sure to understand the results and the discussion about it, can the authors rephrase this section?

Rephrased as suggested.

1) I am skeptical about looking at aerosols next to clouds in general: The presence of a cloud means that the conditions are different than where there is clear sky. How can the authors make sure that the aerosols, meteorological parameters are the same between clear sky and cloudy sky?

In order to obtain 'co-located' aerosol-cloud retrievals for analysis, the often adopted choice is a  $1 \times 1^{\circ}$  gridding, within which sub-grid clear-sky and cloudy pixels co-exist (if clouds are not fully overcast) and are used for retrieving cloud and aerosol properties, respectively. Within the  $1 \times 1^{\circ}$  grid, aerosol concentrations are considered homogeneous (Anderson et al., 2003) so the clear-sky aerosol concentration is representative of that under the cloudy condition. A corresponding statement is added on lines 305-308. Also note that the same meteorological conditions are not required when talking about the representativeness of aerosol, except for relative humidity, which is associated with aerosol swelling. Actually, the swelling effect has been considered by using the metric  $\Delta L$ .

**Added test:**

Lines 305-308: 'In order to obtain horizontally 'co-located' aerosol-cloud retrievals for analysis, the often adopted choice is a 1° by 1° gridding scale, at which aerosol concentrations are considered homogeneous (Anderson et al., 2003). Within a 1° by 1° grid box, sub-grid clear-sky and cloudy pixels co-exist (if clouds are not fully overcast) and are used for retrieving cloud and aerosol properties, respectively.'

2) Also, the authors mentioned that studies on 3D effect and aerosol swelling next to observed clouds are lacking but there are two references about this subject that I think are relevant to this subject and do not appear on the present article: 1. https://doi.org/10.5194/acp-17-13151-2017 2. https://doi.org/10.5194/acp-17-13165-2017

Thanks for these two important references, which are now added. We also rephrase the associated sentences (lines 300-303).

**Added/Modified text:**

Lines 300-303: 'Both aerosol retrievals errors due to 3D radiative effects, cloud contamination, and aerosol swelling, and cloud retrieval errors for 3-D-shaped and heterogeneous clouds, have been shown to artificially introduce biases in the estimation of aerosol-cloud interactions (Quaas et al., 2010; Christensen et al., 2017; Neubauer et al., 2017; Jia et al., 2019, 2021). '

10. Line 264 "It is also noted that SAOD shows first an increase and then decreases from the second DELTA L bin", I do not see what the authors are refereeing to, or is it from the third bin?

We thank the reviewer for the attentive reading of the text. Yes, we meant the third bin. This is now corrected.

11. Line 276 "highly depends on the retrievals bias in clouds", since there is no information on method biases, I believe this statement is too strong.

Thanks for the comment. The method biases have been already inferred by adding the 95% confidence interval as suggested by the reviewer.

12. Line 307, "The strength of the Twomey effect derived on a basis of column-integrated aerosol quantity...", I am confused because on line 294, the authors said that SO4C AND SO4S mimic the column integrated, and here only SO4C shows a large slope so how is it directly linked to this? and SO4S does not show the trend described on line 307.

In this study, we use SO4C to mimic the column integrated aerosols (e.g., AI/AOD) but SO4S to mimic the surface aerosol quantities (e.g., aerosol extinction coefficient). This sentence has been rephrased to avoid misleading.

13. Along the article, different methods are employed to explain the different biases, I was expecting at the end a value of S considering all the possible biases, (precipitation, too close to the cloud, ...) for latitude bands for example and/or season, but no. Each paragraph is developed individually and at the end the discussion does not bring all of them together.

This is a very good suggestion by the reviewer. See response to specific point above(MajorRevision#8).

- 14. Figure 1:
  - Why the blue and white lines do not go until AI=1. The authors mention that the lowest values of AI are removed, but they do not refer to large values of AI (AI>0.5).
     Thanks for the reminder. We did not remove the large values of AI. This is just a plotting thing, and has

Thanks for the reminder. We did not remove the large values of Al. This is just a plotting thing, and has been corrected.

2) Also, in the plot of PDF Vs AI, I do not understand why the PDF is almost equal to 0 for AI 0.5 whereas on the upper plot the PDF of the data are clearly greater than the 0 (might even be higher than the maximum at AI 0.8).

As noted in the original manuscript (the caption of Fig. 1), the upper plot is a joint  $N_{\rm d}$ -AI histogram, where each column is normalized so that it sums to 1. That is, this plot shows the probability of finding a specific  $N_{\rm d}$ , given that a certain AI has been observed, instead of the occurrence frequency distribution of all data.

3) I think it would be better to indicate the value of the regression to see the difference between the blue and white lines and discuss about it, there is nothing about it in the data and method section.

Thanks for the suggestion. The regression slopes (with 95% confidence interval) are now added to the plot and discussed (lines 203-205), and data used to generate this schematic diagram are also clarified in the caption.

Added/Modified text:

Lines 203-205: 'Figure 1 illustrates that the pre-binned approach has a larger slope than lumping together all data points by 18 %, suggesting that attention should be paid when comparing S derived from different approaches. '

4) Why the study uses the blue line instead of the white lines? The authors could infer the uncertainty through the 95% confidence interval using the white lines.

Added. See response to specific point above (MajorRevision#2.5).

**Minor revisions**

1. All along the text, there are several words which are unnecessary in my opinion (e.g., line 112 "basically", line 175 "It is clear that", line 175 "evident", line 183 "remarkably", line 219 "as expected", line 220 "much", line 226 "evidently", line 229 "obviously", line 232 "clearly", line 271 "serious", line 275 "sharply", line 293 "practically", line 319 "much", line 341 "clearly"), and sometime they imply that something is evident but it is not the case (in my opinion).

We thank the reviewer for the attentive reading. Modified as suggested.

2. Line 7 "consistent with stronger aerosol-cloud interactions at larger updraft velocity", this is not the case for every type of clouds (e.g., arctic stratus).

Thanks for the reminder. Corrected.

3. Line 25 "This study will", I suggest moving this sentence to the last sentence of the paragraph.

Thanks for the suggestion. Revised.

4. Line 30 "As reviewed recently...", I do not understand in which context the present study is related to Quaas et al. (2020), will they consider the same biases, new ones... I think, the authors could clarify this part in the introduction. It makes sense afterward but not reading the introduction for the first time.

Thanks for the suggestion. Quaas et al. (2020) is a recent review paper that summarized current challenges and issues obscuring an accurate estimate of the Twomey effect from satellite observations. Building on this review, we investigate several understudied aspects, which are important but not yet understood in a clear way as reviewed by (Quaas et al., 2020). We have added a sentence to make this clear (lines 57-58).

Added text:

Lines 57-58: 'However, a clear understanding on how they affect the estimates of S quantitatively is lacking from the perspective of satellite observations (Quaas et al., 2020). '

5. Line 55 "their strong correlation illustrated by in-situ observations of cumuliform clouds", this sentence is important as it is a key correlation used through the study, so I think the authors could elaborate a bit more on the limitations of it.

Thanks for the comment. We write a statement on this now in the revised manuscript (on lines 68-71).

Added text:

Lines 68-71: 'Although data used to draw this conclusion by Zheng and Rosenfeld (2015) were collected from only three locations, they covered various boundary conditions over both continent and ocean. Moreover, a theoretical framework has also been established to support the observed empirical relationship (Zheng, 2019), lending credibility to applying CBH as a proxy of the updraft.'

6. Line 116 "a standard deviation higher than the mean value", does this threshold come from somewhere specific?

The threshold stems from the study by Saponaro et al. (2017), which has been cited in the main text.

7. Line 117 "the lowest 15%", is this threshold based on Ma et al. (2018) or is it an ad-hoc choice. If the latter, the authors could specify why 15% is chosen.

See response to specific point above (MajorRevision#2.4).

- 8. Lin 123 (Feingold et al., 2021), it should not have parenthesis. Thank you. Corrected.
- 9. Line 140, is there a reason why "MYD06 5-km" is not written "MYD05 5x5 km2" to be consistent with "CloudSat data at a 1.4x2.5km2".

Thanks for the reminder. Corrected.

10. Line 146, the authors mention that they considered sulfate from MERRA-2, I am wondering if they considering other species.

Thanks for the comment. Other species are not considered in this study, because variability in sulfate aerosols has been found to contributes the most strongly to variability in  $N_{\rm d}$  among all aerosol species (McCoy et al., 2017). We now clarify this on lines 185-187.

Added text:

Lines 185-187: 'Given that variability in sulfate aerosols contributes the most strongly to variability in  $N_d$  among all aerosol species (McCoy et al., 2017), the sulfate concentration is considered be to the CCN proxy here.'

11. Line 162 "Nd is essentially a function of both CCN and updraft", a citation should be added here. Thank you. Added on line 210.

Modified text:

Line 210: 'In adiabatic clouds,  $N_d$  is essentially a function of both CCN and updraft (Feingold et al., 2001).'

12. Line 190 "Sai is consistently higher than SAOD", not always as for CGT 900m

We thank the reviewer for this attentive reading of the text. Indeed, it is not the case for CGT 900m. We now revise the text accordingly.

13. Line 195 "They proposed...", Who are they? Are they Reutter and al.? If so, they should not be put in parenthesis in the sentence before.

Yes, 'they' are Reutter et al. Thanks for the comment. Corrected.

14. Line 199 "proxy of updraft (CBH/CGT)", are the quartiles enough to discriminate the updraft regimes described by Reutter et al.? The quartiles defined regimes based on how likely an updraft regime occurs defining different regimes, but I am not sure that they separate in the regimes described by Reutter et al., I am not sure we are in category b.

To check the robustness of the assumption, we further constrain the variation of CGT to a smaller range (i.e., < 10th and > 90th percentiles), and very similar results are obtained (Fig. S2). With the quasi-constant CGT, AI can thus be assumed as an indicator of regime. We now add a statement on this (lines 248).

Added text:

Line 248: 'Note that applying the 10th and 90th percentiles also yields similar results as shown in Fig. S2.'

- 15. Lines 201-203, "As illustrated in the ...." I am confused by this sentence; can the authors rephrase this sentence? What I understand:
  - 1) At low AI, the updraft should have limited impact on Nd (case a from Reutter et al.), but looking at the plot 3c and i, the updraft has a strong impact.

Yes, at low AI, what should be expected is the similar distribution of  $N_d$  between different cloud dynamics as determined by the nature of aerosol-limited regime, or at least a slightly higher  $N_d$  for the strong updraft case. However, looking at the clean zone (AI < 0.15) in Fig. 3, the strong updraft is associated with much lower  $N_d$  as well as larger CER (generally larger than 14 µm, the threshold for drizzle initiation) compared to the weak updraft. Note that this is not relevant to the role of strong updraft facilitating activation of cloud droplets but a possible role of precipitation and/or drizzle. This has been clarified in the original manuscript (now lines 258-263)

2) On the opposite, at high AI, the updraft should most likely have a strong impact, but looking at the plot, the values are messier, and I do not observe a strong dependence on Nd with AI here. Can the dependence be quantified by the authors?

It is known that the impact of aerosol on cloud is strong at low aerosol concentration, whereas a saturation effect occurs as the aerosol keeps rising. Thus the dependence of Nd on AI is supposed to be weak at high AI as we see.

3) Maybe it is irrelevant, but I am wondering why the authors based their regimes on CGT and not the ratio CGT/AI?

Thanks for the question. Since our focus is to contrast  $AI-N_d$  relationships at strong (high) and weak (low) updrafts (CGT), it is more straightforward to illustrate  $AI-N_d$  joint histogram for constrained CGT intervals. By doing so, the information of regimes can be also inferred via AI simply.

16. Line 228 "appears to strengthen the aerosol-Nd relationship", can the author develop a bit more on this?

This is a speculation made by (Painemal et al., 2020) to explain their observed higher aerosol- $N_{\rm d}$  correlation for all clouds compared to non-drizzling clouds. Unfortunately, they did not dig more deeply.

17. Section 3 "AOD", the authors mentioned earlier in the text that they would not consider AOD and prefer the use of AI, but they use AOD in this section. Why changing the considered parameter?

Thank you for this question. We know that AI is not directly retrieved but calculated from AOD, so it is necessary to look at AOD as well when discussing retrieval problems. We now make this clear on lines 321-322.

**Added text:**

Lines 321-322: 'In this section, we also look at AOD in addition to AI, since AOD is a directly retrieved quantity and thus more closely related to retrieval problems.'

18. Lines 264 to 268, I do not understand this part, can the authors rephrase it?

Rephrased as suggested (lines 331-337).

It is also noted that  $S_{AOD}$  ( $S_{AI}$ ) shows an increase first and then a decrease from the third  $\Delta L$  bin. However, the following decrease is unlikely linked to the aerosol retrieval bias since the AOD (AI) remains almost constant (the upper panel in Fig. 5a). One interpretation for this would be that AOD/AI is getting less representative for the aerosol concentrations near cloud with increasing  $\Delta L$ , especially for grid-boxes with precipitation where aerosol is not as homogeneous as assumed (Anderson et al., 2003). Moreover, as  $\Delta L$ is also negatively correlated with CF (Várnai and Marshak, 2015), the decreasing  $S_{AOD}$  ( $S_{AI}$ ) is probably associated to other factors modulated by CF (such as retrieval error in  $N_d$  as demonstrated in the following analysis).'

19. Line 282 "it is clearly illustrated that CF regulates the negative correlation between DELTAL and DELTANd", can the authors provide more information on that part?

The detail on how CF regulates the negative correlation have been added on lines 354-356.

Added text:

Lines 354-356: 'Under the condition of large CF, clear pixels are very close to the nearest cloud pixel, corresponding to a lower  $\Delta L$ , meanwhile, most of sub-grid cloud pixels meet the criteria for confident cloud retrievals, leading to a higher (near-zero)  $\Delta N_d$ ; and the reverse is true in the case of low CF.'

20. Line 287 "Given that CF also correlates closely with cloud dynamics", the correlation presented by the authors are based on the medians which are not very significant in my opinion, 2D histogram and regression on the entire dataset would be better.

Thanks for the comment. The joint histogram between CF and CGT is now added to Fig. 6 as suggested. Given that the relationship is highly non-linear, a single regression slope would not make much sense; such slope is thus not included.

21. Line 295 "commonly used", can the author support this with citations? Maybe some references on the use of satellite observation combined with models to study aerosol cloud interactions are missing here.

"Commonly used" here refers to AOD/AI and surface aerosol extinction coefficient from satellite-based and ground-based observations rather than SO4C and SO4S from reanalysis data. Although not as commonly adopted as AOD/AI and surface aerosol extinction coefficient, SO4C and SO4S were also used as CCN proxies by previous studies (McCoy et al., 2017; Jia et al., 2021). This sentence has been revised to avoid misleading, and the relevant references have been cited as well.

22. Line 296 "is considered to be more relevant to the amount of CCNs", can the author provide a citation for this statement?

Thank you. Added on line 371.

Modified text:

Line 371: 'As demonstrated by Stier (2016), the SO4B derived in combination with CBH, is expected to be more relevant to the amount of CCNs actually activated at cloud base than SO4C and SO4S. '

23. Line 302 "pre-binned method", is it the method described in Figure 1? If this is the case, can it be explicit? I am still skeptical, and I would prefer statistics on the entire dataset and not on the medians (as done in Table S1).

Clarified. As replied above, the analyses on both pre-binned data and entire data are included in the revised manuscript.

24. Line 302 "binend"-> "binned".

Corrected.

25. Line 320 "such as western North Pacific and the Atlantic", this is true for SO4B but not for SO4C (also East coast of south America and South Africa), or maybe I am misunderstanding.

The reviewer is right that high CVs are also evinced at East coast of south America and South Africa. We revise the text accordingly (line 397).

Modified text:

Line 397: 'Spatially, the larger CVs are generally located over the aerosol outflow regions, such as the western North Pacific, the Atlantic, and the east coasts of south America and South Africa, indicative of an impact of the strong variation of continental, and specifically anthropogenic emissions.'

26. Line 321 "the spatial CV of SO4C exhibits a much smaller (0.88) value than those of SO4B and SO4S (1.84 and 1.79)", are these values averaged over North Pacific and Atlantic, if this is the case the authors should specify the limit in lat/lon of the considered box.

The values of spatial CVs are calculated from the global geographical distribution instead regional one. This is now clarified on line 394.

Modified text:

Line 394: 'the spatial CV is derived from the multi-annual averaged global geographical distribution.'

27. Line 330 "loose correlation (R < 0.3) ...", can the authors quantify or rephrase that because I am not convinced especially for r(SO4B, SO4S) which seems high when the ratio is small.

Thanks for the suggestion. We now rephrase this sentence as suggested on line 408. Also note that this sentence is only relevant to r(SO4B,SO4C) instead of r(SO4B,SO4S).

Modified text:

Line 408: 'Interestingly, there is also a good consistency between the spatial patterns of the ratio of SO4BC to SO4C and the correlation coefficient of SO4C with SO4B (Fig. 8d,e), i.e., the high-ratio regions (the ratio > 15 %) generally have strong correlations (R > 0.7). '

28. From line 331 to line 339, I find the conclusions on this discussion very strong. I think it should be at least quantified to support that.

Modified as suggested.

- 29. Line 406 "In terms of aerosol...", is this sentence part of point 3 or point 4? This is a part of point #3. Here, the use of aerosol reanalysis is recommended to avoid retrieval biases in AOD/AI, while in the point 4, the use of reanalysis is to address the problem of vertical co-location.
- 30. Line 419 "SO4B-S-C", can the authors specify again on the different quantity in the conclusions?

Thanks for the reminder. The meanings of SO4C, SO4B, and SO4S have been specified in the conclusion section on lines 506, 507, and 513.

**References**

[revised manuscript text omitted]

---

## Author Response (AR3)

**Response to the Reviewer's Comments**

May 2, 2022

**Comments by Reviewer #2**

I would like to thank the authors for the changes made in the article and the answers to my comments. Reading the new version of the article, I am confident that the changes improved the results, the readability, and the understanding for the readers. I confirm that the study provided a deep analysis on the aerosol-cloud interactions, the results are very interesting and they are suitable for ACP. Nevertheless, I still have some concerns from the analysis and the answer to the comments, therefore I still recommend major revisions before publication in ACP. I detailed the reasons below. I encourage the authors to consider my comments because the results are very interesting and would be useful for the community.

We thank the reviewer for this thoughtful and constructive statement. We have revised the manuscript carefully according to the reviewer's comments. Please see the following detailed point-by-point responses.

**Major revisions**

1. I still disagree on the fact that what is inferred from the study is the Twomey effect (even considering the microphysical part of it). I still believe that calculating the sensitivity of N Vs CCN is the aerosol-cloud interaction but not the Twomey effect (same comment would go for studies looking at re Vs CCN constraining for LWP). Therefore, I find the wording confusing and particularly the title. I advise the authors to keep reference to Twomey as a motivation only. McComiskey and Feingold (2012) made explicit that : "... the terminology for this calculation was changed to ACI (aerosol-cloud interaction) to clarify that the result represents not the indirect effect, which is a response of cloud albedo to aerosol, but instead the microphysical response of the albedo"..

   Thanks for the comment. According to the reviewer's suggestion, we change the 'the Twomey effect' to 'droplet number response to aerosol'. The title is now rephrased as 'Addressing the difficulties in quantifying droplet number response to aerosol from satellite observations'

2. l.192 : The all-data method has one more degree of averaging compared to the pre-binned. Also the 95% confidence interval is also lower with the all data-method. I understand the point about comparing the methods but I do not understand why the pre-binned is preferred in the rest of the study.

   Thanks for the comment. Given that both methods were employed extensively by previous studies and the choice is somewhat ambiguous, here we would not really place a preference. Thus, we show both results (one in the main text and the other in the supplement information) in the revised manuscript, which lead to similar conclusions. The reason why we put the all-data method in the supplement information is that the statistical significance tests become rather meaningless when using the very large amounts of data points.

3. l. 300 : About the aerosol measurements next to clouds, the authors mentioned that the uncertainties potentially come from 3D effects, swelling etc. They did a nice analysis with the DeltaL parameter. Nevertheless I think the study could have gone a bit further with concrete quantifications of the uncertainties. It might be difficult for some parameters but for example RH might be possible to infer from reanalysis. Did the authors look at the uncertainty related to DeltaL constraining for RH or directly related to RH ?

   The reviewer has a great point. We agree that relative humidity (RH) is an important factor to be considered since it can affect Nd via entrainment mixing and also AOD/AI via aerosol swelling. We also thank the suggestion of using RH from reanalysis; actually we have done such analysis in a previous work (Jia et al., 2019) but for CER-to-AI sensitivity, and found a more negative CER-AI slope (equivalent to larger S here) for higher RH condition, which was attributed to the weaker entrainment mixing. Thus, the evidently lower S

in the first DeltaL bin, where data are tightly linked to large CF (Fig. 6), in turn, associated with relatively high RH (Engström and Ekman, 2010), is unlikely to be caused by the effect of RH on clouds.

Considering enormous subgrid-scale variability of RH and un-avoidable co-location issues between reanalysis and satellite observations, it is difficult to obtain precise and coincident RH from reanalysis to match the $\Delta L$ observations. Instead, we choose to utilize the published in-situ aircraft measurements to roughly isolate the contribution of aerosol swelling to the greatly high AOD in the fisrt $\Delta L$ bin from retrieval issues (i.e., 3D radiative effects and cloud contamination). During the Indian Ocean Experiment (INDOEX), Twohy et al. (2009) measured a rise in RH from about 70% at more than 20 km from cloud to 90% with 1–4 km of cloud edge (equivalent to the distances of the third and first $\Delta L$ bins in Fig. 5a), which in turn results in about a 69% increase in aerosol scattering cross section (Twohy et al., 2009). Considering the aerosol humidification only occurs near cloud level, i.e., one quarter to one third of the aerosol column could be affected according to Lidar observations (Twohy et al., 2009), the increase in AOD by aerosol swelling is estimated to be 17–23%. This is up to about a third of relative increase in AOD from the third to first $\Delta L$ bins (64%) in Fig. 5a, implying that the retrieval errors in aerosol contribute the majority of the S reduction in the first $\Delta L$ bin. Note that the estimated AOD rise due to humidification relies on observed RH variability surrounding cloud and also the vertical profile and chemical composition of aerosol. The associated discussion is now added on lines 313-323.

*Added/Modified text:*

*Lines 313-323: 'Based on the published in-situ aircraft measurements, we roughly isolate the contribution of aerosol swelling from retrieval issues (i.e., 3D radiative effects and cloud contamination). During the Indian Ocean Experiment (INDOEX), Twohy et al. (2009) measured a rise in relative humidity (RH) from about 70% at more than 20 km from cloud to 90% with 1–4 km of cloud edge (equivalent to the distances of the third and first $\Delta L$ bins in Fig. 5a), which in turn results in about a 69% increase in aerosol scattering cross section (Twohy et al., 2009). Considering the aerosol humidification only occurs near cloud level, i.e., one quarter to one third of the aerosol column could be affected according to Lidar observations (Twohy et al., 2009), the increase in AOD by aerosol swelling is estimated to be 17–23%. This is up to about a third of relative increase in AOD from the third to first $\Delta L$ bins (64%) in Fig. 5a, implying that the retrieval errors in aerosol could contribute the majority of the S reduction in the first $\Delta L$ bin. It should be noted that the estimated AOD rise due to humidification relies on observed RH variability surrounding cloud and also the vertical profile and chemical composition of aerosol, which could vary with geographic location.*

**Minor revisions**

1. l. 137-140 : The study from Anderson et al. (2003) considers clear-sky pixels but they do not conclude anything when it is cloudy (I think). I still do not know if considering aerosols next to clouds represent aerosols within the clouds. There might be a reason why cloudy pixels are cloudy and adjacent clear-sky pixels, clear sky. Not everything can be uniform, maybe it comes from the aerosols. I do not expect the authors to redo the analysis with another proxy (but I would recommend it for their future analysis) but I would like the text to highlight this concern clearly.

   Thanks for the suggestion. We agree with the reviewer that aerosol is not always as homogeneous as assumed, especially when precipitation occurs. We now clarify this on lines 135-136.

   *Added text:*

   *Lines 135-136: ' Note that this assumption would be questionable especially when aerosol is scavenged by precipitation (Gryspeerdt et al., 2015).'*

2. About the uncertainties, I was referring to the ones derived from the algorithms directly, there is no quantification about the uncertainty from Nd or CER for example. The authors constrained on the latitudes but the uncertainties on the retrieved parameters can still be high..

   Thanks for the comment. The discussion on retrieval uncertainties is now added on lines 155-158.

   *Added/Modified text:*

   *Lines 155-158: 'With the above sampling strategy, the random uncertainty in $N_d$ was reported at 78% on a pixel level and this dropped substantially when averaged to a 1° by 1° region (Grosvenor et al., 2018).*

*However, as stated in Gryspeerdt et al. (2021), the systematic bias in the $N_d$ retrievals to in situ measurements is low, with determination coefficients of 0.48 for all cloud types and 0.5-0.8 for stratocumulus clouds. '*

3. l. 149-151 : "Nd id the independent variable in the S calculation", I am not sure to understand what is meant here, can the author rephrase?.

   Here we meant that the adiabaticity doesn't affect the sensitivity calculation if it is constant or doesn't significantly vary with AOD (AI). Thanks for the reminder. We find this sentence is not necessary, and remove it now.

4. Table 3 is interesting but I am wondering how the authors determined the magnitude of the biases for each issue. Also would it be possible to retrieve a percentage of the bias (even a rough estimate) for each issue to estimate the uncertainties on S? If a future study cannot account for an effect, it would be very useful to use this information..

   We thanks the reviewer for this thoughtful suggestion. This is a great point! We now update Table 3 with the percentage of each bias as suggested.

   *Added/Modified text:*

   *Lines 486-492: ' Aerosol retrieval biases (3D radiative effects and cloud contamination), aerosol swelling, and cloud retrieval bias (heterogeneity effect) tend to lead to an underestimation of S. Although $S_{AI}$ ($S_{AOD}$) for the first $\Delta L$ bin, where evident AI(AOD) enhancement exists, is about 29% (50%) less than other unaffected bins, the overall underestimation is only ∼3% because of the small data volume in the first bin (Fig. 5a). Nevertheless, for low-$\Delta L$ dominated regions (e.g., stratcumulus regions), the underestimation can be more pronounced. By comparing $S_{AI}$ ($S_{AOD}$) calculated by $N_{dAll}$ and $N_d$, the underestimation by cloud retrieval issues is roughly estimated to be ∼8% (∼17%). '*

**References**

Engström, A. and Ekman, A. M. L.: Impact of meteorological factors on the correlation between aerosol optical depth and cloud fraction, Res. Lett, 37, 18 814, https://doi.org/10.1029/2010GL044361, 2010.

Grosvenor, D. P., Sourdeval, O., Zuidema, P., Ackerman, A., Alexandrov, M. D., Bennartz, R., Boers, R., Cairns, B., Chiu, J. C., Christensen, M., Deneke, H., Diamond, M., Feingold, G., Fridlind, A., Hünerbein, A., Knist, C., Kollias, P., Marshak, A., McCoy, D., Merk, D., Painemal, D., Rausch, J., Rosenfeld, D., Russchenberg, H., Seifert, P., Sinclair, K., Stier, P., van Diedenhoven, B., Wendisch, M., Werner, F., Wood, R., Zhang, Z., and Quaas, J.: Remote Sensing of Droplet Number Concentration in Warm Clouds: A Review of the Current State of Knowledge and Perspectives, Rev. Geophys., 56, 409–453, https://doi.org/10.1029/2017RG000593, 2018.

Gryspeerdt, E., Stier, P., White, B. A., and Kipling, Z.: Wet scavenging limits the detection of aerosol effects on precipitation, Atmos. Chem. Phys., 15, 7557–7570, https://doi.org/10.5194/acp-15-7557-2015, 2015.

Gryspeerdt, E., McCoy, D. T., Crosbie, E., Moore, R. H., Nott, G. J., Painemal, D., Small-Griswold, J., Sorooshian, A., and Ziemba, L.: The impact of sampling strategy on the cloud droplet number concentration estimated from satellite data, Atmos. Meas. Tech. Discuss., 2021, 1–25, https://doi.org/10.5194/amt-2021-371, 2021.

Jia, H., Ma, X., Quaas, J., Yin, Y., and Qiu, T.: Is positive correlation between cloud droplet effective radius and aerosol optical depth over land due to retrieval artifacts or real physical processes?, Atmos. Chem. Phys., https://doi.org/10.5194/acp-19-8879-2019, 2019.

Twohy, C. H., Coakley, J. A., and Tahnk, W. R.: Effect of changes in relative humidity on aerosol scattering near clouds, J. Geophys. Res, 114, 5205, https://doi.org/10.1029/2008JD010991, 2009.